# Adaptive Variance Reduction for Stochastic Optimization under Weaker Assumptions

**Wei Jiang**[1], **Sifan Yang**[1,2], **Yibo Wang**[1,2], **Lijun Zhang**[1,2,*]

[1]National Key Laboratory for Novel Software Technology, Nanjing University, Nanjing, China

[2]School of Artificial Intelligence, Nanjing University, Nanjing, China

{jiangw, yangsf, wangyb, zhanglj}@lamda.nju.edu.cn

## Abstract

This paper explores adaptive variance reduction methods for stochastic optimization based on the STORM technique. Existing adaptive extensions of STORM rely on strong assumptions like bounded gradients and bounded function values, or suffer an additional $\mathcal{O}(\log T)$ term in the convergence rate. To address these limitations, we introduce a novel adaptive STORM method that achieves an optimal convergence rate of $\mathcal{O}(T^{-1/3})$ for non-convex functions with our newly designed learning rate strategy. Compared with existing approaches, our method requires weaker assumptions and attains the optimal convergence rate without the additional $\mathcal{O}(\log T)$ term. We also extend the proposed technique to stochastic compositional optimization, obtaining the same optimal rate of $\mathcal{O}(T^{-1/3})$. Furthermore, we investigate the non-convex finite-sum problem and develop another innovative adaptive variance reduction method that achieves an optimal convergence rate of $\mathcal{O}(n^{1/4}T^{-1/2})$, where $n$ represents the number of component functions. Numerical experiments across various tasks validate the effectiveness of our method.

## 1 Introduction

This paper investigates the stochastic optimization problem

$$\min_{\mathbf{x} \in \mathbb{R}^d} f(\mathbf{x}), \tag{1}$$

where $f : \mathbb{R}^d \mapsto \mathbb{R}$ is a smooth non-convex function. We assume that only noisy estimations of its gradient $\nabla f(\mathbf{x})$ can be accessed, denoted as $\nabla f(\mathbf{x}; \xi)$, where $\xi$ represents the random sample drawn from a stochastic oracle such that $\mathbb{E}[\nabla f(\mathbf{x}; \xi)] = \nabla f(\mathbf{x})$.

Problem (1) has been comprehensively investigated in the literature [Duchi et al., 2011, Kingma and Ba, 2015, Loshchilov and Hutter, 2017], and it is well-known that the classical stochastic gradient descent (SGD) achieves a convergence rate of $\mathcal{O}(T^{-1/4})$, where $T$ denotes the iteration number [Ghadimi and Lan, 2013]. To further improve the convergence rate, variance reduction methods have been developed, and attain an improved rate of $\mathcal{O}(T^{-1/3})$ under a slightly stronger smoothness assumption [Fang et al., 2018, Wang et al., 2019]. However, these methods necessitate the use of a huge batch size in each iteration, which is often impractical to use. To eliminate the need for large batches, a momentum-based variance reduction method — STORM [Cutkosky and Orabona, 2019] is introduced, which achieves a convergence rate of $\mathcal{O}(T^{-1/3} \log T)$.

Although aforementioned methods are equipped with convergence guarantees, their analyses rely on delicate configurations of hyper-parameters, such as the learning rate and the momentum parameter. To set them properly, the algorithm typically needs to know the value of the smoothness parameter $L$,

---

*Lijun Zhang is the corresponding author.

the gradient upper bound $G$, and the variance upper bound $\sigma$, which are often unknown in practice. Specifically, most algorithms require the learning rate $\eta_t$ smaller than $\mathcal{O}(1/L)$, and for the STORM method, setting the momentum parameter to $\mathcal{O}(L^2\eta_t^2)$ is crucial for ensuring convergence [Cutkosky and Orabona, 2019].

To overcome this limitation, many adaptive algorithms have been developed, aiming to obtain convergence guarantees without prior knowledge of problem-dependent parameters such as $L$, $G$ and $\sigma$. Based on the STORM method, Levy et al. [2021] develop the STORM+ algorithm, which attains the optimal $\mathcal{O}(T^{-1/3})$ convergence rate under the assumption of bounded function values and gradients. To remove the need for the bounded function values assumption, Liu et al. [2022] propose the META-STORM algorithm to attain an $\mathcal{O}(T^{-1/3} \log T)$ convergence rate, but it still requires the bounded gradients assumption and includes an additional $\mathcal{O}(\log T)$ term. In summary, despite advancements in this field, existing adaptive STORM-based methods either depend on strong assumptions or suffer an extra $\mathcal{O}(\log T)$ term compared with the lower bound [Arjevani et al., 2023]. Hence, a fundamental question to be addressed is:

> *Is it possible to develop an adaptive STORM method that achieves the optimal convergence rate for non-convex functions under weaker assumptions?*

We give an affirmative answer to the above question by devising a novel optimal Adaptive STORM method (Ada-STORM). The learning rate of our algorithm is set to be inversely proportional to a specific power of the iteration number $T$ in the initial iterations, and then changes adaptively based on the cumulative sum of past gradient estimations. In this way, we are able to adjust the learning rate dynamically according to the property of stochastic gradients, and ensure a small learning rate in the beginning. Leveraging this strategy, Ada-STORM achieves an optimal convergence rate of $\mathcal{O}(T^{-1/3})$ for non-convex functions. Notably, our analysis does not require the function to have bounded values and bounded gradients, which is a significant advancement over existing methods [Levy et al., 2021, Liu et al., 2022]. Additionally, our convergence rate does not contain the extra $\mathcal{O}(\log T)$ term, which is often present in STORM-based methods [Cutkosky and Orabona, 2019, Liu et al., 2022]. To highlight the versatility of our approach and its potential impact in the field of stochastic optimization, we further extend our technique to develop optimal adaptive methods for compositional optimization.

Finally, we investigate adaptive variance reduction for the non-convex finite-sum problems. Inspired by SAG algorithm [Roux et al., 2012], we incorporate an additional term in the STORM estimator, which measures the difference of past gradients between the selected component function and the overall objective. By changing the learning rate according to the sum of past gradient estimations, we are able to obtain an optimal convergence rate of $\mathcal{O}(n^{-1/4}T^{-1/2})$ for finite-sum problems in an adaptive manner, where $n$ is the number of component functions. Our result is better than the previous convergence rate of $\mathcal{O}(n^{-1/4}T^{-1/2} \log(nT))$ obtained by adaptive SPIDER method [Kavis et al., 2022]. In summary, compared with existing methods, this paper enjoys the following advantages:

- For stochastic non-convex optimization, our method achieves the optimal convergence rate of $\mathcal{O}(T^{-1/3})$ under more relaxed assumptions. Specifically, it does not require the bounded function values or the bounded gradients, and does not include the additional $\mathcal{O}(\log T)$ term in the convergence rate.
- Our learning rate design and the analysis exhibit broad applicability. We substantiate this claim by obtaining an optimal rate of $\mathcal{O}(T^{-1/3})$ for stochastic compositional optimization, using the technique proposed in this paper.
- For non-convex finite-sum optimization, we further improve our adaptive algorithm to attain an optimal convergence rate of $\mathcal{O}(n^{1/4}T^{-1/2})$, which outperforms the previous result by eliminating the $\mathcal{O}(\log(nT))$ factor.

A comparison between our method and other STORM-based algorithms is shown in Table 1. Numerical experiments on different tasks also validate the effectiveness of the proposed method.

## 2 Related work

This section briefly reviews related work on stochastic variance reduction methods and adaptive stochastic algorithms.

Table 1: Summary of results for STORM-based methods. Here, NC denotes non-convex, Comp. indicates compositional optimization, FS represents finite-sum optimization, and BG/BF refers to requiring bounded gradients or bounded function values assumptions. Adaptive means the method does not require to know problem-dependent parameters, i.e., $L$, $G$, and $\sigma$.

| Method | Setting | Convergence Rate | Adaptive | BG/BF |
|--------|---------|------------------|----------|-------|
| STORM [Cutkosky and Orabona, 2019] | NC | $\mathcal{O}\left(T^{-1/3}\log T\right)$ | ✗ | ✓ |
| Super-ADAM [Huang et al., 2021] | NC | $\mathcal{O}\left(T^{-1/3}\log T\right)$ | ✗ | – |
| STORM+ [Levy et al., 2021] | NC | $\mathcal{O}\left(T^{-1/3}\right)$ | ✓ | ✓ |
| META-STORM [Liu et al., 2022] | NC | $\mathcal{O}\left(T^{-1/3}\log T\right)$ | ✓ | ✓ |
| **Theorem 1, 2** | NC | $\mathcal{O}\left(T^{-1/3}\right)$ | ✓ | – |
| **Theorem 3** | NC & Comp. | $\mathcal{O}\left(T^{-1/3}\right)$ | ✓ | – |
| **Theorem 4** | NC & FS | $\mathcal{O}\left(n^{1/4}T^{-1/2}\right)$ | ✓ | – |

## 2.1 Stochastic variance reduction methods

Variance reduction has been widely used in stochastic optimization to reduce the gradient estimation error and thus improve the convergence rates. The idea of variance reduction can be traced back to the SAG algorithm [Roux et al., 2012], which incorporates a memory of previous gradient values to ensure variance reduction and achieves a linear convergence rate for strongly convex finite-sum optimization. To avoid the storage of past gradients, SVRG [Zhang et al., 2013, Johnson and Zhang, 2013] proposes to calculate the full gradient periodically, obtaining the same convergence rate as the SAG algorithm. Subsequent advancement has been made by the SARAH method [Nguyen et al., 2017], which derives better convergence for smooth convex functions.

In the context of non-convex objectives, Fang et al. [2018] introduce the SPIDER estimator, which improves the convergence rate from $\mathcal{O}(T^{-1/4})$ to $\mathcal{O}(T^{-1/3})$ in stochastic settings, and to $O(n^{1/4}T^{-1/2})$ in finite-sum scenarios, with $n$ representing the number of components in the finite-sum. Following this, the SpiderBoost algorithm [Wang et al., 2019] refines the SPIDER approach by employing a larger constant step size and adapting it for composite optimization problems. However, a common limitation among these methods is their reliance on large batch sizes for each iteration, posing practical challenges due to high computational demands. To mitigate this issue, Cutkosky and Orabona [2019] introduce the STORM method, a momentum-based technique that achieves an $\mathcal{O}(T^{-1/3}\log T)$ convergence rate without using large batches. Concurrently, Tran-Dinh et al. [2019] obtain the same result using a similar algorithm but through a different analysis.

## 2.2 Adaptive stochastic algorithms

For stochastic optimization, it is well-known that the SGD algorithm can obtain a convergence rate of $\mathcal{O}(T^{-1/4})$ for non-convex objective functions with well-designed learning rates [Ghadimi and Lan, 2013]. Instead of using pre-defined iteration-based learning rates, many stochastic methods propose to adjust the learning rate based on past stochastic gradients. One of the foundational works is the AdaGrad algorithm [Duchi et al., 2011], which proves to be effective for sparse data. Further advancements include RMSprop [Tieleman and Hinton, 2012] and Adam [Kingma and Ba, 2015], demonstrating broad effectiveness across a wide range of machine learning problems. Later, the Super-Adam [Huang et al., 2021] algorithm further improves the Adam algorithm via the variance reduction technique STORM [Cutkosky and Orabona, 2019] and obtains a convergence rate of $\mathcal{O}(T^{-1/3}\log T)$. Nevertheless, to obtain the corresponding convergence rates, these methods still require knowledge of certain problem-dependent parameters to set hyper-parameters accurately, hence

**Algorithm 1** STORM Algorithm

---
1: **Input:** time step $T$, initial point $\mathbf{x}_1$
2: **for** time step $t = 1$ **to** $T$ **do**
3:    Set hyper-parameters $\beta_t$ and $\eta_t$
4:    Compute $\mathbf{v}_t$ according to equation (2)
5:    Update the decision variable: $\mathbf{x}_{t+1} = \mathbf{x}_t - \eta_t \mathbf{v}_t$
6: **end for**
7: Choose $\tau$ uniformly at random from $\{1, \ldots, T\}$
8: Return $\mathbf{x}_\tau$

---

not adaptive.[2] To solve this problem, many research aims to develop fully adaptive SGD methods that maintain the optimal convergence rate without knowing problem-specific parameters [Orabona, 2014, Chen et al., 2022, Carmon and Hinder, 2022, Ivgi et al., 2023, Yang et al., 2023].

Recently, adaptive adaptations of STORM have received considerable attention. A notable development is the introduction of STORM+ [Levy et al., 2021], which presents a fully adaptive version of STORM while attaining an optimal convergence rate. To circumvent the bounded function values assumption in STORM+, the META-STORM [Liu et al., 2022] approach is developed, equipped with a nearly optimal bound. However, META-STORM still requires the bounded gradients assumption, and it includes an additional $\mathcal{O}(\log T)$ in the convergence rate. Consequently, adaptive STORM with the optimal convergence rate and under mild assumptions still needs further explorations.

## 3 Adaptive variance reduction for non-convex optimization

In this section, we develop an adaptive STORM method for non-convex functions. We first outline the assumptions used, and then present our proposed method and analyze its convergence rate.

### 3.1 Assumptions

We introduce the following assumptions, which are standard and commonly adopted in the stochastic optimization [Nguyen et al., 2017, Fang et al., 2018, Cutkosky and Orabona, 2019, Li et al., 2021].

**Assumption 1** *(Average smoothness)*

$$\mathbb{E}\left[\|\nabla f(\mathbf{x}; \xi) - \nabla f(\mathbf{y}; \xi)\|^2\right] \leq L^2 \|\mathbf{x} - \mathbf{y}\|^2.$$

**Assumption 2** *(Bounded variance)*

$$\mathbb{E}\left[\|\nabla f(\mathbf{x}; \xi) - \nabla f(\mathbf{x})\|^2\right] \leq \sigma^2.$$

**Assumption 3** $f_* = \inf_{\mathbf{x}} f(\mathbf{x}) \geq -\infty$ *and* $f(\mathbf{x}_1) - f_* \leq \Delta_f$ *for the initial solution* $\mathbf{x}_1$.

Note that some additional assumptions are required in other STORM-based methods. Specifically, STORM [Cutkosky and Orabona, 2019], STORM+ [Levy et al., 2021], and META-STORM [Liu et al., 2022] assume the bounded gradients. Moreover, STORM+ makes an additional assumption of the bounded function values.

### 3.2 The proposed method

In this subsection, we aim to develop an adaptive STORM method that achieves an optimal convergence rate for non-convex functions under weaker assumptions. Our algorithm framework is the same as the original STORM [Cutkosky and Orabona, 2019], and the only difference is the setup of the momentum parameter $\beta_t$ and the learning rate $\eta_t$. First, we present the STORM algorithm in Algorithm 1.

---
[2]In this paper, adaptive means the algorithm does not require problem-dependent parameters to set up hyper-parameters such as the learning rate and the momentum parameter.

The core idea of STORM lies in a carefully devised variance reduced estimator $\mathbf{v}_t$, which effectively tracks the gradient $\nabla f(\mathbf{x}_t)$. For the first iteration ($t = 1$), we set $\mathbf{v}_1 = \sum_{i=1}^{B_0} \frac{1}{B_0} \nabla f(\mathbf{x}_1; \xi_1^i)$, which is estimated within a batch $B_0 = T^{1/3}$. Note that we use large batch only in the first iteration, and constant batch size in other iterations. In subsequent iterations ($t \geq 2$), estimator $\mathbf{v}_t$ is defined as:

$$\mathbf{v}_t = (1 - \beta_t)\mathbf{v}_{t-1} + \beta_t \nabla f(\mathbf{x}_t; \xi_t) + (1 - \beta_t)\left(\nabla f(\mathbf{x}_t; \xi_t) - \nabla f(\mathbf{x}_{t-1}; \xi_t)\right), \quad (2)$$

where the first two terms are similar to the momentum SGD, and the last term serves as the error correction, which ensures the variance reduction effect. By choosing the values of $\beta_t$ and $\eta_t$ carefully, STORM ensures that the estimation error $\mathbb{E}[\|\mathbf{v}_t - \nabla f(\mathbf{x}_t)\|^2]$ would decrease gradually. In the original STORM paper, these parameters are set up as:

$$\eta_t = \frac{k}{\left(w + \sum_{i=1}^t \|\nabla f(\mathbf{x}_t; \xi_t)\|^2\right)^{1/3}}, \quad \beta_t = c\eta_t^2,$$

where $k = \mathcal{O}(G^{2/3}L^{-1})$, $w = \mathcal{O}(G^2)$ and $c = \mathcal{O}(L^2)$. The settings of these hyper-parameters are crucial to the convergence analysis of STORM. However, it is worth noting that $L$ is the smoothness parameter and $G$ is the gradient upper bound, which are often difficult to determine in practice.

To address this problem, our approach defines the hyper-parameters as follows:

$$\eta_t = \min\left\{\frac{1}{T^{1/3}}, \frac{1}{T^{(1-\alpha)/3}\left(\sum_{i=1}^t \|\mathbf{v}_i\|^2\right)^\alpha}\right\}, \quad \beta_t = \beta = T^{-2/3}, \quad (3)$$

where $0 < \alpha < 1/3$. Notably, our method does not rely on the parameters $L$ and $G$, and also does not need the bounded gradients or bounded function values assumptions that are common in other methods. Although our formulation initially requires knowledge of the iteration number $T$, this can be effectively circumvented using the doubling trick, which will be explained later. The above learning rate $\eta_t$ can also be expressed in an alternative, more illustrative manner:

$$\eta_t = \begin{cases} \frac{1}{T^{1/3}} & \text{if } \sum_{i=1}^t \|\mathbf{v}_i\|^2 \leq T^{1/3}; \\ \frac{1}{T^{(1-\alpha)/3}\left(\sum_{i=1}^t \|\mathbf{v}_i\|^2\right)^\alpha} & \text{else.} \end{cases}$$

This formulation ensures that the learning rate starts sufficiently small in the initial stages and then changes dynamically based on the gradient estimator $\mathbf{v}_t$. This design makes our learning rate setup and convergence analysis distinctly different from previous methods. Next, we present the following theoretical guarantee for our algorithm.

**Theorem 1** *Under Assumptions 1, 2 and 3, Algorithm 1 with hyper-parameters in equation (3) guarantees that:*

$$\mathbb{E}\left[\|\nabla f(\mathbf{x}_\tau)\|\right] \leq \mathcal{O}\left(\frac{\Delta_f^{\frac{1}{2(1-\alpha)}} + \sigma^{\frac{1}{1-\alpha}} + L^{\frac{1}{2\alpha}}}{T^{1/3}}\right).$$

**Remark:** To ensure that $\mathbb{E}[\|\nabla f(\mathbf{x}_\tau)\|] \leq \epsilon$, the overall complexity is $\mathcal{O}(\epsilon^{-3})$, which is known to be optimal up to constant factors [Arjevani et al., 2023]. Compared with existing STORM-based algorithms [Cutkosky and Orabona, 2019, Levy et al., 2021, Liu et al., 2022], our method does not have the extra $\mathcal{O}(\log T)$ term in the convergence rate, and our analysis does not require bounded gradients or bounded function values assumptions. Also note that the selection of $\alpha$ does not affect the order of $T$, and larger $\alpha$ leads to better dependence on parameter $L$ and worse reliance on parameters $\Delta$ and $\sigma$. Considering we require that $0 < \alpha < 1/3$, we can simply set $\alpha = 0.3$ in practice.

## 3.3 The doubling trick

While we have attained the optimal convergence rate using the proposed adaptive STORM method, it requires knowing the total number of iterations $T$ in advance. Here, we show that we can avoid this requirement by using the doubling trick, which divides the algorithm into several stages and increases

the iteration number in each stage gradually. Specifically, we design a multi-stage algorithm over $k = \{1, 2, \cdots, K\}$ stages. At the beginning of each new stage, we reset $\mathbf{x}_t = \mathbf{x}_0$. In each stage $k$, the STORM algorithm is executed for $2^{k-1}$ iterations, effectively doubling the iteration numbers after each stage. In any step $t$, we first identify the current stage as $1 + \lfloor \log t \rfloor$ and then calculate the iteration number for this stage as $I_t = 2^{\lfloor \log t \rfloor}$. Then, we can set the hyper-parameters as:

$$\eta_t = \min \left\{ \frac{1}{I_t^{1/3}}, \frac{1}{I_t^{(1-\alpha)/3} \left( \sum_{i=I_t}^t \|\mathbf{v}_i\|^2 \right)^\alpha} \right\}, \quad \beta_t = I_t^{-2/3}, \quad I_t = 2^{\lfloor \log t \rfloor}. \tag{4}$$

This approach eliminates the need to predetermine the iteration number $T$. By using the doubling trick, we can still obtain the same optimal convergence rate as stated in the following theorem.

**Theorem 2** *Under Assumptions 1, 2 and 3, Algorithm 1 with hyper-parameters in equation (4) guarantees that:*

$$\mathbb{E}\left[\|\nabla f(\mathbf{x}_\tau)\|\right] \leq \mathcal{O}\left( \frac{\Delta_f^{\frac{1}{2(1-\alpha)}} + \sigma^{\frac{1}{1-\alpha}} + L^{\frac{1}{2\alpha}}}{T^{1/3}} \right).$$

## 4 Extension to stochastic compositional optimization

To demonstrate the broad applicability of our proposed technique, we extend it to stochastic compositional optimization [Wang et al., 2017a,b, Yuan et al., 2019, Zhang and Xiao, 2019, 2021, Jiang et al., 2023, 2024a], formulated as:

$$\min_{\mathbf{x} \in \mathbb{R}^d} F(\mathbf{x}) = f(g(\mathbf{x})), \tag{5}$$

where $f$ and $g$ are smooth functions. We assume that we can only access to unbiased estimations of $\nabla f(\mathbf{x})$, $\nabla g(\mathbf{x})$ and $g(\mathbf{x})$, denoted as $\nabla f(\mathbf{x}; \xi)$, $\nabla g(\mathbf{x}; \zeta)$ and $g(\mathbf{x}; \zeta)$. Here $\xi$ and $\zeta$ symbolize the random sample drawn for a stochastic oracle such that $\mathbb{E}[\nabla f(\mathbf{x}; \xi)] = \nabla f(\mathbf{x})$, $\mathbb{E}[g(\mathbf{x}; \zeta)] = g(\mathbf{x})$, and $\mathbb{E}[\nabla g(\mathbf{x}; \zeta)] = \nabla g(\mathbf{x})$.

Existing variance reduction methods [Hu et al., 2019, Zhang and Xiao, 2019, Qi et al., 2021] are able to obtain optimal $\mathcal{O}(T^{-1/3})$ convergence rates for problem (5), but they require the knowledge of smoothness parameter and the gradient upper bound to set up hyper-parameters. In this section, we aim to achieve the same optimal convergence rate without prior knowledge of problem-dependent parameters. We develop our adaptive algorithm for this problem as follows. In each step $t$, the algorithm maintains an inner function estimator $\mathbf{u}_t$ in the style of STORM, i.e.,

$$\mathbf{u}_t = (1 - \beta)\mathbf{u}_{t-1} + g(\mathbf{x}_t; \zeta_t) - (1 - \beta)g(\mathbf{x}_{t-1}; \zeta_t). \tag{6}$$

Then, we construct a gradient estimator $\mathbf{v}_t$ based on $\mathbf{u}_t$ also in the style of STORM:

$$\mathbf{v}_t = (1 - \beta)\mathbf{v}_{t-1} + \nabla f(\mathbf{u}_t; \xi_t)\nabla g(\mathbf{x}_t; \zeta_t) - (1 - \beta)\nabla f(\mathbf{u}_{t-1}; \xi_t)\nabla g(\mathbf{x}_{t-1}; \zeta_t). \tag{7}$$

After that, we apply gradient descent using the gradient estimator $\mathbf{v}_t$. The whole algorithm is presented in Algorithm 2, and hyper-parameters are set the same as in equation (3). For the first iteration, we simply set $\mathbf{u}_1 = \sum_{i=1}^{B_0} \frac{1}{B_0} g(\mathbf{x}_1; \zeta_1^i)$ and $\mathbf{v}_1 = \sum_{i=1}^{B_0} \frac{1}{B_0} \nabla f(\mathbf{u}_1; \xi_1^i)\nabla g(\mathbf{x}_1; \zeta_1)$, where $B_0 = T^{1/3}$. Next, we list common assumptions used in the literature of compositional optimization [Wang et al., 2017a,b, Yuan et al., 2019, Zhang and Xiao, 2019, 2021].

**Assumption 4** *(Average smoothness and Lipschitz continuity)*

$$\mathbb{E}\left[\|\nabla f(\mathbf{x}; \xi) - \nabla f(\mathbf{y}; \xi)\|^2\right] \leq L\|\mathbf{x} - \mathbf{y}\|^2; \ \mathbb{E}\left[\|f(\mathbf{x}; \xi) - f(\mathbf{y}; \xi)\|^2\right] \leq C\|\mathbf{x} - \mathbf{y}\|^2;$$

$$\mathbb{E}\left[\|\nabla g(\mathbf{x}; \zeta) - \nabla g(\mathbf{y}; \zeta)\|^2\right] \leq L\|\mathbf{x} - \mathbf{y}\|^2; \ \mathbb{E}\left[\|g(\mathbf{x}; \zeta) - g(\mathbf{y}; \zeta)\|^2\right] \leq C\|\mathbf{x} - \mathbf{y}\|^2.$$

**Assumption 5** *(Bounded variance)*

$$\mathbb{E}\left[\|g(\mathbf{x}; \zeta) - g(\mathbf{x})\|^2\right] \leq \sigma^2; \mathbb{E}\left[\|\nabla g(\mathbf{x}; \zeta) - \nabla g(\mathbf{x})\|^2\right] \leq \sigma^2; \mathbb{E}\left[\|\nabla f(\mathbf{x}; \xi) - \nabla f(\mathbf{x})\|^2\right] \leq \sigma^2.$$

---
**Algorithm 2** Compositional STORM
---
1: **Input:** time step $T$, initial point $\mathbf{x}_1$
2: **for** time step $t = 1$ **to** $T$ **do**
3:     Compute $\mathbf{u}_t$ according to equation (6)
4:     Compute $\mathbf{v}_t$ according to equation (7)
5:     Update the decision variable: $\mathbf{x}_{t+1} = \mathbf{x}_t - \eta_t \mathbf{v}_t$
6: **end for**
7: Choose $\tau$ uniformly at random from $\{1, \ldots, T\}$
8: Return $\mathbf{x}_\tau$
---

**Assumption 6** $F_* = \inf_{\mathbf{x}} F(\mathbf{x}) \geq -\infty$ *and* $F(\mathbf{x}_1) - F_* \leq \Delta_F$ *for the initial solution* $\mathbf{x}_1$.

**Remark:** In Assumption 4, we further require standard Lipschitz continuity assumption, which is essential and widely required in the literature for stochastic compositional optimization [Wang et al., 2017b, Yuan et al., 2019, Jiang et al., 2022a,b]. This assumption is inherently introduced by the compositional optimization itself rather than by our adaptive techniques.

With the above assumptions, our algorithm enjoys the following guarantee.

**Theorem 3** *Under Assumptions 4, 5 and 6, our Algorithm 2 ensures that:*

$$\mathbb{E}\left[\|\nabla F(\mathbf{x}_\tau)\|\right] \leq \mathcal{O}\left(T^{-1/3}\right).$$

**Remark:** This rate matches the state-of-the-art (SOTA) results in stochastic compositional optimization [Hu et al., 2019, Zhang and Xiao, 2019, Qi et al., 2021], and our method achieve this in an adaptive manner. Note that our convergence rate aligns with the lower bound for single-level problems [Arjevani et al., 2023] and is thus unimprovable.

## 5 Adaptive variance reduction for finite-sum optimization

In this section, we further improve our adaptive variance reduction method to obtain an enhanced convergence rate for non-convex finite-sum optimization, which is in the form of

$$\min_{\mathbf{x} \in \mathbb{R}^d} F(\mathbf{x}) = \frac{1}{n} \sum_{i=1}^{n} f_i(\mathbf{x}),$$

where each $f_i(\cdot)$ is a smooth non-convex function. Existing adaptive method for this problem [Kavis et al., 2022] achieves a convergence rate of $\mathcal{O}(n^{1/4}T^{-1/2}\log(nT))$ based on the variance reduction technique SPIDER [Fang et al., 2018], suffering from an extra $\mathcal{O}(\log(nT))$ term compared with the corresponding lower bound [Fang et al., 2018, Li et al., 2021].

To obtain the optimal convergence rate for finite-sum optimization, we incorporate techniques from the SAG algorithm [Roux et al., 2012] into the STORM estimator. Specifically, in each step $t$, we start by randomly sample $i_t$ from the set $\{1, 2, \cdots, n\}$. Then, we construct a variance reduction gradient estimator as

$$\mathbf{v}_t = (1 - \beta)\mathbf{v}_{t-1} + \nabla f_{i_t}(\mathbf{x}_t) - (1 - \beta)\nabla f_{i_t}(\mathbf{x}_{t-1}) - \beta\left(g_t^{i_t} - \frac{1}{n}\sum_{i=1}^{n} g_t^i\right), \quad (8)$$

where the first three terms align with the original STORM method, and the last term, inspired by the SAG algorithm, deals with the finite-sum structure. Here, $g_t$ tracks the gradient as

$$g_{t+1}^i = \begin{cases} \nabla f_{i_t}(\mathbf{x}_t) & i = i_t \\ g_t^i & i \neq i_t \end{cases}. \quad (9)$$

By such a design, we can ensure that the estimation error $\mathbb{E}[\|\mathbf{v}_t - \nabla F(\mathbf{x}_t)\|^2]$ reduces gradually. The whole algorithm is stated in Algorithm 3. In this case, we set the hyper-parameters as:

$$\eta_t = \frac{1}{n^{\frac{1-\alpha}{2}}\left(\sum_{i=1}^{t} \|\mathbf{v}_i\|^2\right)^\alpha}, \quad \beta = \frac{1}{n},$$

---

**Algorithm 3** STORM for Finite-sum Optimization (SAG-type)

---
1: **Input:** time step $T$, initial point $\mathbf{x}_1$
2: **for** time step $t = 1$ **to** $T$ **do**
3:     Sample $i_t$ randomly from $\{1, 2, \cdots, n\}$
4:     Compute estimator $\mathbf{v}_t$ according to equation (8)
5:     Update $g_{t+1}$ according to equation (9)
6:     Update the decision variable: $\mathbf{x}_{t+1} = \mathbf{x}_t - \eta_t \mathbf{v}_t$
7: **end for**
8: Choose $\tau$ uniformly at random from $\{1, \ldots, T\}$
9: Return $\mathbf{x}_\tau$

---

where $0 < \alpha < 1/3$. The learning rate $\eta_t$ is non-increasing and changes according to the gradient estimations, and the momentum parameter $\beta$ remains unchanged throughout the learning process. Next, we show that our method enjoys the optimal convergence rate under the following assumptions, which are standard and widely adopted in existing literature [Fang et al., 2018, Wang et al., 2019, Li et al., 2021].

**Assumption 7** *(Smoothness) For each $i \in \{1, 2, \cdots, m\}$, function $f_i$ is $L$-smooth such that*

$$\|\nabla f_i(\mathbf{x}) - \nabla f_i(\mathbf{y})\| \leq L\|\mathbf{x} - \mathbf{y}\|.$$

**Assumption 8** $F_* = \inf_{\mathbf{x}} F(\mathbf{x}) \geq -\infty$ *and* $F(\mathbf{x}_1) - F_* \leq \Delta_F$ *for the initial solution* $\mathbf{x}_1$.

**Theorem 4** *Under Assumptions 7 and 8, our Algorithm 3 guarantees that:*

$$\mathbb{E}\left[\|\nabla F(\mathbf{x}_\tau)\|\right] \leq \mathcal{O}\left(\frac{n^{1/4}}{T^{1/2}}\left(\Delta_F^{\frac{1}{2(1-\alpha)}} + L^{\frac{1}{2\alpha}}\right)\right).$$

**Remark:** Our result matches the lower bound for non-convex finite-sum problems [Fang et al., 2018, Li et al., 2021], and makes an improvement over the existing adaptive method, i.e., AdaSpider [Kavis et al., 2022]. Specifically, the convergence rate of the AdaSpider algorithm is $\mathcal{O}\left(n^{1/4}T^{-1/2}\left(L^2 + \Delta_F\right) \cdot \log\left(1 + nTL\right)\right)$, and our result is better than theirs when $\frac{1}{4} < \alpha < \frac{1}{3}$.

We can avoid storing past gradients by following the SVRG method [Zhang et al., 2013, Johnson and Zhang, 2013] to compute the full gradient periodically and incorporate it into STORM estimator. Instead of storing the past gradients as in SAG algorithm, we can avoid this storage cost by incorporating elements from the SVRG method. Specifically, we compute a full batch gradient at the first step and every $I$ iteration (we set $I = n$):

$$\nabla f(\mathbf{x}_\tau) = \frac{1}{n}\sum_{i=1}^{n}\nabla f_i(\mathbf{x}_\tau).$$

For other iterations, we randomly select an index $i_t$ from the set $\{1, 2, \cdots, n\}$ and compute:

$$\mathbf{v}_t = (1-\beta)\mathbf{v}_{t-1} + \nabla f_{i_t}(\mathbf{x}_t) - (1-\beta)\nabla f_{i_t}(\mathbf{x}_{t-1}) - \beta\left(\nabla f_{i_t}(\mathbf{x}_\tau) - \nabla f(\mathbf{x}_\tau)\right). \tag{10}$$

Note that the first three terms match the original STORM estimator, and the last term, inspired from SVRG, deals with the finite-sum structure. Compared with equation (8) in Algorithm 3, the difference is that we use $\left(\nabla f_{i_t}(\mathbf{x}_\tau) - \nabla f(\mathbf{x}_\tau)\right)$ instead of $\left(g_t^{i_t} - \frac{1}{n}\sum_{i=1}^{n}g_t^i\right)$ in the last term. The detailed procedure is outlined in Algorithm 4. This strategy maintains the same optimal rate, as stated below:

**Theorem 5** *Under Assumptions 7 and 8, our Algorithm 4 guarantees that:*

$$\mathbb{E}\left[\|\nabla F(\mathbf{x}_\tau)\|\right] \leq \mathcal{O}\left(\frac{n^{1/4}}{T^{1/2}}\left(\Delta_F^{\frac{1}{2(1-\alpha)}} + L^{\frac{1}{2\alpha}}\right)\right).$$

**Remark:** The obtained convergence rate is in the same order as the results in Theorem 4, and Algorithm 4 does not require storing past gradients anymore.

**Algorithm 4** STORM for Finite-sum Optimization (SVRG-type)

1: **Input:** time step $T$, initial point $\mathbf{x}_1$
2: **for** time step $t = 1$ **to** $T$ **do**
3:    **if** $t \mod I == 0$ **then**
4:       Set $t = \tau$ and compute $\nabla f(\mathbf{x}_\tau) = \frac{1}{n} \sum_{i=1}^n \nabla f_i(\mathbf{x}_\tau)$
5:    **end if**
6:    Sample $i_t$ randomly from $\{1, 2, \cdots, n\}$
7:    Compute $\mathbf{v}_t$ according to equation (10)
8:    Update the decision variable: $\mathbf{x}_{t+1} = \mathbf{x}_t - \eta \mathbf{v}_t$
9: **end for**
10: Select $\tau$ uniformly at random from $\{1, \ldots, T\}$
11: Return $\mathbf{x}_\tau$

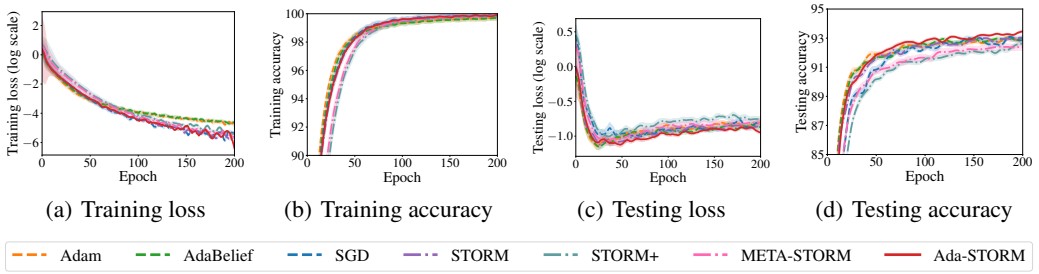

(a) Training loss    (b) Training accuracy    (c) Testing loss    (d) Testing accuracy

Figure 1: Results for CIFAR-10 dataset.

## 6 Experiments

In this section, we evaluate the performance of the proposed Ada-STORM method via numerical experiments on image classification tasks and language modeling tasks. In the experiments, we compare our method with STORM [Cutkosky and Orabona, 2019], STORM+ [Levy et al., 2021] and META-STORM [Liu et al., 2022], as well as SGD, Adam [Kingma and Ba, 2015] and AdaBelief [Zhuang et al., 2020]. We use the default implementation of SGD and Adam from Pytorch [Paszke et al., 2019]. For STORM+, we follow its original implementation[3], and build STORM, META-STORM and our Ada-STORM based on it. When it comes to hyper-parameter tuning, we simply set $\alpha = 0.3$ for our algorithm. For other methods, we either set the hyper-parameters as recommended in the original papers or tune them by grid search. For example, we search the learning rate of SGD, Adam and AdaBelief from the set $\{1e-5, 1e-4, 1e-3, 1e-2, 1e-1\}$ and select the best one for each method. All the experiments are conducted on eight NVIDIA Tesla V100 GPUs.

**Image classification task**    First, we conduct numerical experiments on multi-class image classification tasks to evaluate the performance of the proposed method. Specifically, we train ResNet18 and ResNet34 models [He et al., 2016] on the CIFAR-10 and CIFAR-100 datasets [Krizhevsky, 2009] respectively. For all optimizers, we set the batch size as 256 and train for 200 epochs. We plot the loss value and the accuracy against the epochs on the CIFAR-10 and CIFAR-100 datasets in Figure 1 and Figure 2. It is observed that, for training loss and training accuracy, our Ada-STORM algorithm achieves comparable performance with respect to other methods, and it outperforms the others in terms of testing loss and thus obtains a better testing accuracy.

**Language modeling task**    Then, we perform experiments on language modeling tasks. Concretely, we train a 2-layer Transformer [Vaswani et al., 2017] over the WiKi-Text2 dataset [Merity, 2016]. We use 256 dimensional word embeddings, 512 hidden unites and 2 heads. The batch size is set as 20 and all methods are trained for 40 epochs with dropout rate 0.1. We also clip the gradients by norm 0.25 in case of the exploding gradient. We report both the loss and perplexity versus the number of epochs in Figure 3. From the results, we observe that our method converges more quickly than other

---

[3]https://github.com/LIONS-EPFL/storm-plus-code

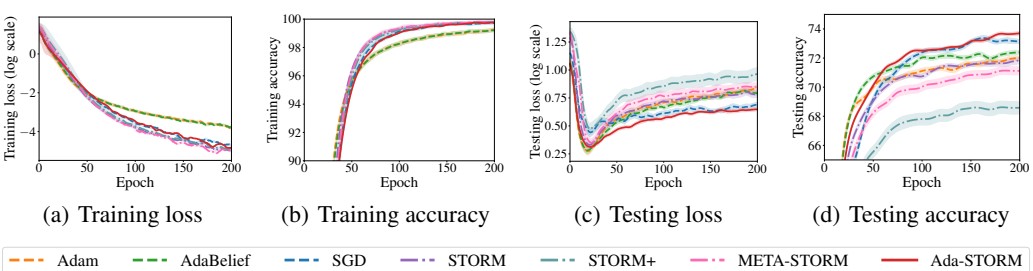

(a) Training loss  (b) Training accuracy  (c) Testing loss  (d) Testing accuracy

Figure 2: Results for CIFAR-100 dataset.

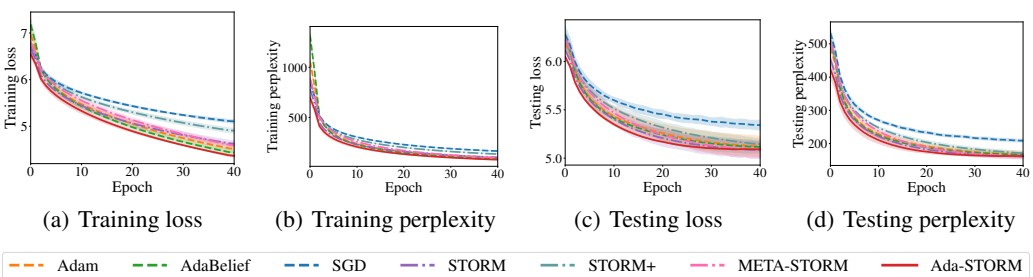

(a) Training loss  (b) Training perplexity  (c) Testing loss  (d) Testing perplexity

Figure 3: Results for WikiText-2 dataset.

methods and obtains a slightly better perplexity compared with others, indicating the effectiveness of the proposed method.

# 7  Conclusion

In this paper, we propose an adaptive STORM method to achieve the optimal convergence rate for non-convex functions. Compared with existing methods, our algorithm requires weaker assumptions and does not have the additional $\mathcal{O}(\log T)$ term in the convergence rate. The proposed technique can also be employed to develop optimal adaptive algorithms for compositional optimization. Furthermore, we investigate an adaptive method for non-convex finite-sum optimization, obtaining an improved convergence rate of $\mathcal{O}(n^{1/4}T^{-1/2})$. Given that STORM algorithm has already been used in many areas such as bi-level optimization [Yang et al., 2021], federated learning [Das et al., 2022], min-max optimization [Xian et al., 2021], sign-based optimization [Jiang et al., 2024b], etc., the proposed methods may also inspire the development of adaptive algorithms in these fields.

# Acknowledgements

This work was partially supported by National Key R&D Program of China (2021ZD0112802), NSFC (62122037), and the Postgraduate Research & Practice Innovation Program of Jiangsu Province (No. KYCX24_0231).

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

# A  Proof of Theorem 1

First, we introduce the following lemma, which is frequently used in our proof.

**Lemma 1** *Suppose $c_i$ is positive for $i = \{1, 2, \cdots, n\}$, and let $0 < \alpha < 1$. We can ensure that:*

$$\left( \sum_{i=1}^{n} c_i \right)^{1-\alpha} \leq \sum_{i=1}^{n} \frac{c_i}{\left( \sum_{j=1}^{i} c_j \right)^{\alpha}} \leq \frac{1}{1-\alpha} \left( \sum_{i=1}^{n} c_i \right)^{1-\alpha}.$$

**Proof 1** *The proof mainly follows McMahan and Streeter [2010] and a similar analysis also appears in Levy et al. [2021]. First, we prove the right part by Induction.*

*(1) For $n = 1$, we can easily show that the hypothesis holds:*

$$\frac{c_1}{c_1^{\alpha}} = c_1^{1-\alpha} \leq \frac{1}{1-\alpha} c_1^{1-\alpha}.$$

*(2) Next, assuming that the hypothesis holds for $n = t-1$, then we show that it also holds for $n = t$. Define $Z = \sum_{i=1}^{t} c_i$ and $X = c_t$. For $n = t$, we have*

$$\sum_{i=1}^{t} \frac{c_i}{\left( \sum_{j=1}^{i} c_j \right)^{\alpha}} = \sum_{i=1}^{t-1} \frac{c_i}{\left( \sum_{j=1}^{i} c_j \right)^{\alpha}} + \frac{c_t}{\left( \sum_{j=1}^{t} c_j \right)^{\alpha}}$$

$$\leq \frac{1}{1-\alpha} \left( \sum_{i=1}^{t-1} c_i \right)^{1-\alpha} + \frac{c_t}{\left( \sum_{j=1}^{t} c_j \right)^{\alpha}}$$

$$= \frac{1}{1-\alpha} (Z - X)^{1-\alpha} + \frac{X}{Z^{\alpha}} := h(X).$$

*Taking the derivative concerning $x$, we know that*

$$\frac{dh(X)}{dX} = \frac{1}{Z^{\alpha}} - \frac{1}{(Z-X)^{\alpha}},$$

*which indicates that $h(X)$ decreases as $X$ increasing. Since $0 \leq X \leq Z$,*

$$\max_{0 \leq X \leq Z} h(X) = h(0) = \frac{1}{1-\alpha} Z^{1-\alpha} = \frac{1}{1-\alpha} \left( \sum_{i=1}^{t} c_i \right)^{1-\alpha},$$

*which implies that the hypothesis is true for $n = t$.*

*Combining (1) and (2), we finish the proof for the right part. Then, we give the proof of the left part as follows:*

$$\sum_{i=1}^{n} \frac{c_i}{\left( \sum_{j=1}^{i} c_j \right)^{\alpha}} \geq \sum_{i=1}^{n} \frac{c_i}{\left( \sum_{j=1}^{n} c_j \right)^{\alpha}} = \left( \sum_{i=1}^{n} c_i \right)^{1-\alpha}.$$

*Thus, we finish the proof of this lemma.*

Next, we can obtain the following guarantee for our algorithm.

**Lemma 2** *Our method enjoys the following guarantee:*

$$\sum_{t=1}^{T} \mathbb{E} \left[ \eta_t \, \|\mathbf{v}_t\|^2 \right]$$

$$\leq \left( 2\Delta_f + \sigma^2 \right) + \mathbb{E} \underbrace{\left[ \frac{2L^2}{\beta} \sum_{t=1}^{T} \eta_t^3 \, \|\mathbf{v}_t\|^2 \right]}_{(A)} + \mathbb{E} \underbrace{\left[ L \sum_{t=1}^{T} \eta_t^2 \, \|\mathbf{v}_t\|^2 \right]}_{(B)} + \mathbb{E} \underbrace{\left[ 2\beta\sigma^2 \sum_{t=1}^{T} \eta_t \right]}_{(C)}.$$

**Proof 2** *According to the definition of estimator* $\mathbf{v}_t$, *we can deduce that:*

$$
\begin{aligned}
&\mathbb{E}_{\xi_{t+1}}\left[\|\nabla f(\mathbf{x}_{t+1}) - \mathbf{v}_{t+1}\|^2\right] \\
&= \mathbb{E}_{\xi_{t+1}}\left[\|(1-\beta)\mathbf{v}_t + \nabla f(\mathbf{x}_{t+1};\xi_{t+1}) - (1-\beta)\nabla f(\mathbf{x}_t;\xi_{t+1}) - \nabla f(\mathbf{x}_{t+1})\|^2\right] \\
&= \mathbb{E}_{\xi_{t+1}}[\|(1-\beta)(\mathbf{v}_t - \nabla f(\mathbf{x}_t)) + (\nabla f(\mathbf{x}_t) - \nabla f(\mathbf{x}_{t+1}) + \nabla f(\mathbf{x}_{t+1};\xi_{t+1}) - \nabla f(\mathbf{x}_t;\xi_{t+1})) \\
&\quad + \beta\left(\nabla f(\mathbf{x}_t;\xi_{t+1}) - \nabla f(\mathbf{x}_t)\right)\|^2] \\
&\leq \mathbb{E}_{\xi_{t+1}}\left[(1-\beta)^2\|\mathbf{v}_t - \nabla f(\mathbf{x}_t)\|^2\right] + \mathbb{E}_{\xi_{t+1}}[\|\nabla f(\mathbf{x}_t) - \nabla f(\mathbf{x}_{t+1}) \\
&\quad + \nabla f(\mathbf{x}_{t+1};\xi_{t+1}) - \nabla f(\mathbf{x}_t;\xi_{t+1}) + \beta\left(\nabla f(\mathbf{x}_t;\xi_{t+1}) - \nabla f(\mathbf{x}_t)\right)\|^2] \\
&\leq \mathbb{E}_{\xi_{t+1}}\left[(1-\beta)^2\|\mathbf{v}_t - \nabla f(\mathbf{x}_t)\|^2\right] \\
&\quad + 2\mathbb{E}_{\xi_{t+1}}\left[\|\nabla f(\mathbf{x}_t) - \nabla f(\mathbf{x}_{t+1}) + \nabla f(\mathbf{x}_{t+1};\xi_{t+1}) - \nabla f(\mathbf{x}_t;\xi_{t+1})\|^2\right] \\
&\quad + 2\mathbb{E}_{\xi_{t+1}}\left[\|\beta\left(\nabla f(\mathbf{x}_t;\xi_{t+1}) - \nabla f(\mathbf{x}_t)\right)\|^2\right] \\
&\leq (1-\beta)^2\mathbb{E}_{\xi_{t+1}}\left[\|\mathbf{v}_t - \nabla f(\mathbf{x}_t)\|^2\right] + 2\beta^2\mathbb{E}_{\xi_{t+1}}\left[\|\nabla f(\mathbf{x}_t;\xi_{t+1}) - \nabla f(\mathbf{x}_t)\|^2\right] \\
&\quad + 2\mathbb{E}_{\xi_{t+1}}\left[\|\nabla f(\mathbf{x}_{t+1};\xi_{t+1}) - \nabla f(\mathbf{x}_t;\xi_{t+1})\|^2\right] \\
&\leq (1-\beta)\|\mathbf{v}_t - \nabla f(\mathbf{x}_t)\|^2 + 2\beta^2\sigma^2 + 2L^2\|\mathbf{x}_{t+1} - \mathbf{x}_t\|^2 \\
&= (1-\beta)\|\mathbf{v}_t - \nabla f(\mathbf{x}_t)\|^2 + 2\beta^2\sigma^2 + 2L^2\eta_t^2\|\mathbf{v}_t\|^2.
\end{aligned}
\tag{11}
$$

*Note that* $\eta_t$ *is independent of random variable* $\xi_{t+1}$. *So we can guarantee that:*

$$
\mathbb{E}_{\xi_{t+1}}\left[\eta_t\|\nabla f(\mathbf{x}_{t+1}) - \mathbf{v}_{t+1}\|^2\right] \leq (1-\beta)\eta_t\|\mathbf{v}_t - \nabla f(\mathbf{x}_t)\|^2 + 2\beta^2\sigma^2\eta_t + 2L^2\eta_t^3\|\mathbf{v}_t\|^2.
$$

*After rearranging, we have:*

$$
\begin{aligned}
&\eta_t\|\mathbf{v}_t - \nabla f(\mathbf{x}_t)\|^2 \\
&\leq \frac{\eta_t}{\beta}\|\mathbf{v}_t - \nabla f(\mathbf{x}_t)\|^2 - \mathbb{E}_{\xi_{t+1}}\left[\frac{\eta_t}{\beta}\|\mathbf{v}_{t+1} - \nabla f(\mathbf{x}_{t+1})\|^2\right] + 2\beta\sigma^2\eta_t + \frac{2L^2}{\beta}\eta_t^3\|\mathbf{v}_t\|^2.
\end{aligned}
$$

*Letting* $\mathcal{H}_t$ *be the history to time* $t$, *i.e.,* $\mathcal{H}_t = \{\xi_1, \cdots, \xi_t\}$, *we ensure that*

$$
\begin{aligned}
\mathbb{E}_{\mathcal{H}_t}\left[\eta_t\|\mathbf{v}_t - \nabla f(\mathbf{x}_t)\|^2\right] \leq{} &\mathbb{E}_{\mathcal{H}_t}\left[\frac{\eta_t}{\beta}\|\mathbf{v}_t - \nabla f(\mathbf{x}_t)\|^2\right] - \mathbb{E}_{\mathcal{H}_{t+1}}\left[\frac{\eta_t}{\beta}\|\mathbf{v}_{t+1} - \nabla f(\mathbf{x}_{t+1})\|^2\right] \\
&+ 2\beta\sigma^2\mathbb{E}_{\mathcal{H}_t}\left[\eta_t\right] + \frac{2L^2}{\beta}\mathbb{E}_{\mathcal{H}_t}\left[\eta_t^3\|\mathbf{v}_t\|^2\right].
\end{aligned}
$$

*By summing up and noting that* $\eta_t$ *is non-increasing such that* $\eta_{t+1} \leq \eta_t$, *we have:*

$$
\begin{aligned}
&\sum_{t=1}^{T}\mathbb{E}_{\mathcal{H}_t}\left[\eta_t\|\mathbf{v}_t - \nabla f(\mathbf{x}_t)\|^2\right] \\
&\leq \mathbb{E}_{\mathcal{H}_1}\left[\frac{\eta_1}{\beta}\|\mathbf{v}_1 - \nabla f(\mathbf{x}_1)\|^2\right] + 2\beta\sigma^2\sum_{t=1}^{T}\mathbb{E}_{\mathcal{H}_t}\left[\eta_t\right] + \frac{2L^2}{\beta}\sum_{t=1}^{T}\mathbb{E}_{\mathcal{H}_t}\left[\eta_t^3\|\mathbf{v}_t\|^2\right].
\end{aligned}
\tag{12}
$$

*Since we use a large batch size in the first iteration, that is,* $B_0 = T^{1/3}$, *we can now ensure that* $\mathbb{E}_{\mathcal{H}_1}\left[\|\mathbf{v}_1 - \nabla f(\mathbf{x}_1)\|^2\right] \leq \frac{\sigma^2}{B_0} = \frac{\sigma^2}{T^{1/3}}$. *Due to the fact that* $\eta_1 \leq T^{-1/3}$ *and* $\beta = T^{-2/3}$, *the first term of the above inequality is less than* $\sigma^2$. *So, we can finally have:*

$$
\sum_{t=1}^{T}\mathbb{E}\left[\eta_t\|\mathbf{v}_t - \nabla f(\mathbf{x}_t)\|^2\right] \leq \sigma^2 + 2\beta\sigma^2\sum_{t=1}^{T}\mathbb{E}\left[\eta_t\right] + \frac{2L^2}{\beta}\sum_{t=1}^{T}\mathbb{E}\left[\eta_t^3\|\mathbf{v}_t\|^2\right].
\tag{13}
$$

*Also, due to the smoothness of $f(\mathbf{x})$, we know that:*

$$f(\mathbf{x}_{t+1}) \leq f(\mathbf{x}_t) + \langle \nabla f(\mathbf{x}_t), \mathbf{x}_{t+1} - \mathbf{x}_t \rangle + \frac{L}{2} \|\mathbf{x}_{t+1} - \mathbf{x}_t\|^2$$

$$= f(\mathbf{x}_t) - \eta_t \langle \nabla f(\mathbf{x}_t), \mathbf{v}_t \rangle + \frac{\eta_t^2 L}{2} \|\mathbf{v}_t\|^2$$

$$= f(\mathbf{x}_t) - \eta_t \langle \nabla f(\mathbf{x}_t), \mathbf{v}_t \rangle + \frac{\eta_t}{2} \|\nabla f(\mathbf{x}_t)\|^2 + \frac{\eta_t}{2} \|\mathbf{v}_t\|^2 - \frac{\eta_t}{2} \|\nabla f(\mathbf{x}_t)\|^2$$

$$\quad - \frac{\eta_t}{2} \|\mathbf{v}_t\|^2 + \frac{\eta_t^2 L}{2} \|\mathbf{v}_t\|^2$$

$$= f(\mathbf{x}_t) + \frac{\eta_t}{2} \|\nabla f(\mathbf{x}_t) - \mathbf{v}_t\|^2 - \frac{\eta_t}{2} \|\nabla f(\mathbf{x}_t)\|^2 - \frac{\eta_t}{2} \|\mathbf{v}_t\|^2 + \frac{\eta_t^2 L}{2} \|\mathbf{v}_t\|^2.$$

*By summing up and re-arranging, we have:*

$$\sum_{t=1}^{T} \eta_t \|\mathbf{v}_t\|^2 \leq 2f(\mathbf{x}_1) - 2f(\mathbf{x}_{T+1}) + \sum_{t=1}^{T} \eta_t \|\nabla f(\mathbf{x}_t) - \mathbf{v}_t\|^2 + \sum_{t=1}^{T} \eta_t^2 L \|\mathbf{v}_t\|^2. \quad (14)$$

*Then, by using equation (13) and the fact that $f(\mathbf{x}_1) - f_* \leq \Delta_f$, we have:*

$$\sum_{t=1}^{T} \mathbb{E}\left[\eta_t \|\mathbf{v}_t\|^2\right] \leq 2\Delta_f + \sigma^2 + 2\beta\sigma^2 \sum_{t=1}^{T} \mathbb{E}[\eta_t] + 2L^2 \sum_{t=1}^{T} \mathbb{E}\left[\frac{\eta_t^3}{\beta} \|\mathbf{v}_t\|^2\right] + L \sum_{t=1}^{T} \mathbb{E}\left[\eta_t^2 \|\mathbf{v}_t\|^2\right],$$

*which finishes the proof of Lemma 2.*

To effectively bound each term in the above lemma, we divide the algorithm into two stages. Suppose that starting from iteration $t = s$, the condition $\sum_{i=1}^{t} \|\mathbf{v}_i\|^2 \geq T^{1/3}$ begins to hold. We refer to iterations $t = \{1, 2, \cdots, s-1\}$ as the first stage, and $t = \{s, \cdots, T\}$ as the second stage.

**Bounding LHS:** In the first stage, we know that

$$\sum_{t=1}^{s-1} \eta_t \|\mathbf{v}_t\|^2 = \frac{1}{T^{1/3}} \sum_{t=1}^{s-1} \|\mathbf{v}_t\|^2.$$

For the second stage, our analysis leads to:

$$\sum_{t=s}^{T} \eta_t \|\mathbf{v}_t\|^2 = \sum_{t=s}^{T} \frac{\|\mathbf{v}_t\|^2}{T^{\frac{1-\alpha}{3}} \left(\sum_{i=1}^{t} \|\mathbf{v}_i\|^2\right)^\alpha}$$

$$\geq \sum_{t=s}^{T} \frac{\|\mathbf{v}_t\|^2}{T^{\frac{1-\alpha}{3}} \left(T^{\frac{1}{3}} + \sum_{i=s}^{t} \|\mathbf{v}_i\|^2\right)^\alpha}$$

$$\geq \sum_{t=s}^{T} \frac{\|\mathbf{v}_t\|^2}{T^{\frac{1-\alpha}{3}} \left(T^{\frac{\alpha}{3}} + \left(\sum_{i=s}^{t} \|\mathbf{v}_i\|^2\right)^\alpha\right)}$$

$$\geq \sum_{t=s}^{T} \frac{\|\mathbf{v}_t\|^2}{T^{\frac{1-\alpha}{3}} \cdot 2 \max\left\{T^{\frac{\alpha}{3}}, \left(\sum_{i=s}^{t} \|\mathbf{v}_i\|^2\right)^\alpha\right\}}$$

$$\geq \frac{1}{2T^{\frac{1-\alpha}{3}}} \min\left\{\frac{1}{T^{\frac{\alpha}{3}}} \sum_{t=s}^{T} \|\mathbf{v}_t\|^2, \left(\sum_{t=s}^{T} \|\mathbf{v}_t\|^2\right)^{1-\alpha}\right\}$$

$$= \frac{1}{2} \underbrace{\min\left\{\frac{1}{T^{1/3}} \sum_{t=s}^{T} \|\mathbf{v}_t\|^2, \left(\frac{1}{T^{1/3}} \sum_{t=s}^{T} \|\mathbf{v}_t\|^2\right)^{1-\alpha}\right\}}_{:=\Gamma},$$

where the first inequality stems from $\sum_{t=1}^{s-1} \|\mathbf{v}_t\|^2 \le T^{1/3}$, the second inequality results from $(x+y)^\alpha \le x^\alpha + y^\alpha$ for positive $x, y$ and $0 < \alpha < 1/3$, and the forth inequality applies Lemma 1. Next, we bound $(A), (B), (C)$ as follows.

**Bounding** $(A)$**:** In the first stage, with $\eta_t = T^{-1/3}$, $\beta = T^{-2/3}$, and $\sum_{t=1}^{s-1} \|\mathbf{v}_t\|^2 \le T^{1/3}$, we can derive:

$$\frac{2L^2}{\beta} \sum_{t=1}^{s-1} \eta_t^3 \|\mathbf{v}_t\|^2 = \frac{2L^2}{T^{1/3}} \sum_{t=1}^{s-1} \|\mathbf{v}_t\|^2 \le 2L^2.$$

For the second stage, the analysis gives:

$$\frac{2L^2}{\beta} \sum_{t=s}^{T} \eta_t^3 \|\mathbf{v}_t\|^2 \le 2L^2 \sum_{t=s}^{T} \frac{\|\mathbf{v}_t\|^2}{T^{\frac{1-3\alpha}{3}} \left(\sum_{i=s}^{t} \|\mathbf{v}_i\|^2\right)^{3\alpha}} \le \frac{2L^2}{1-3\alpha} \left(\frac{1}{T^{1/3}} \sum_{t=s}^{T} \|\mathbf{v}_t\|^2\right)^{1-3\alpha},$$

where the second inequality uses Lemma 1. Then we also have:

$$\frac{2L^2}{1-3\alpha} \left(\frac{1}{T^{1/3}} \sum_{t=s}^{T} \|\mathbf{v}_t\|^2\right)^{1-3\alpha} = \frac{2L^2}{1-3\alpha} (8-24\alpha)^{1-3\alpha} \left(\frac{1}{(8-24\alpha)} \frac{1}{T^{1/3}} \sum_{t=s}^{T} \|\mathbf{v}_t\|^2\right)^{1-3\alpha}$$

$$\le 3\alpha \left(\frac{2L^2}{1-3\alpha} (8-24\alpha)^{1-3\alpha}\right)^{\frac{1}{3\alpha}} + \frac{1}{8T^{1/3}} \sum_{t=s}^{T} \|\mathbf{v}_t\|^2,$$

where the inequality employs Young's inequality, such that $xy \le 3\alpha x^{\frac{1}{3\alpha}} + (1-3\alpha) y^{\frac{1}{1-3\alpha}}$ for positive $x, y$. Very similarly, we have:

$$\frac{2L^2}{1-3\alpha} \left(\frac{1}{T^{1/3}} \sum_{t=s}^{T} \|\mathbf{v}_t\|^2\right)^{1-3\alpha}$$

$$= \frac{2L^2}{1-3\alpha} \left(\frac{8-24\alpha}{1-\alpha}\right)^{\frac{1-3\alpha}{1-\alpha}} \left(\frac{1-\alpha}{8-24\alpha}\right)^{\frac{1-3\alpha}{1-\alpha}} \left(\frac{1}{T^{1/3}} \sum_{t=s}^{T} \|\mathbf{v}_t\|^2\right)^{1-3\alpha}$$

$$\le \frac{2\alpha}{1-\alpha} \left(\frac{2L^2}{1-3\alpha} \left(\frac{8-24\alpha}{1-\alpha}\right)^{\frac{1-3\alpha}{1-\alpha}}\right)^{\frac{1-\alpha}{2\alpha}} + \frac{1}{8} \left(\frac{1}{T^{1/3}} \sum_{t=s}^{T} \|\mathbf{v}_t\|^2\right)^{1-\alpha},$$

where the second inequality employs Young's inequality, such that $xy \le \frac{2\alpha}{1-\alpha} x^{\frac{1-\alpha}{2\alpha}} + \frac{1-3\alpha}{1-\alpha} y^{\frac{1-\alpha}{1-3\alpha}}$ for positive $x, y$. Combining all above, we know that

$$(A) \le 2L^2 + 3\alpha \left(\frac{2L^2}{1-3\alpha} (8-24\alpha)^{1-3\alpha}\right)^{\frac{1}{3\alpha}}$$

$$+ \frac{2\alpha}{1-\alpha} \left(\frac{2L^2}{1-3\alpha} \left(\frac{8-24\alpha}{1-\alpha}\right)^{\frac{1-3\alpha}{1-\alpha}}\right)^{\frac{1-\alpha}{2\alpha}} + \frac{\Gamma}{8}.$$

**Bounding** $(B)$**:** In the first stage, with the learning rate set at $\eta_t = T^{-1/3}$ and $\sum_{t=1}^{s-1} \|\mathbf{v}_t\|^2 \le T^{1/3}$, we observe:

$$L \sum_{t=1}^{s-1} \eta_t^2 \|\mathbf{v}_t\|^2 = \frac{L}{T^{2/3}} \sum_{t=1}^{s-1} \|\mathbf{v}_t\|^2 \le \frac{L}{T^{1/3}} \le L.$$

For the second stage, our analysis reveals:

$$L \sum_{t=s}^{T} \eta_t^2 \|\mathbf{v}_t\|^2 \le L \sum_{t=s}^{T} \frac{\|\mathbf{v}_t\|^2}{T^{\frac{2(1-\alpha)}{3}} \left(\sum_{i=s}^{t} \|\mathbf{v}_i\|^2\right)^{2\alpha}} \le \frac{L}{1-2\alpha} \left(\frac{1}{T^{1/3}} \sum_{t=s}^{T} \|\mathbf{v}_t\|^2\right)^{1-2\alpha},$$

where the second inequality leverages Lemma 1, and we have:

$$\frac{L}{1-2\alpha}\left(\frac{1}{T^{1/3}}\sum_{t=s}^{T}\|\mathbf{v}_t\|^2\right)^{1-2\alpha}$$

$$=\frac{L}{1-2\alpha}(8-16\alpha)^{1-2\alpha}\frac{1}{(8-16\alpha)^{1-2\alpha}}\left(\frac{1}{T^{1/3}}\sum_{t=s}^{T}\|\mathbf{v}_t\|^2\right)^{1-2\alpha}$$

$$\leq 2\alpha\left(\frac{(8-16\alpha)^{1-2\alpha}L}{1-2\alpha}\right)^{\frac{1}{2\alpha}}+\frac{1}{8T^{1/3}}\sum_{t=s}^{T}\|\mathbf{v}_t\|^2,$$

where the inequality is due to Young's inequality, such that $xy \leq 2\alpha x^{\frac{1}{2\alpha}}+(1-2\alpha)y^{\frac{1}{1-2\alpha}}$ for positive $x, y$. Very similarly, we have:

$$\frac{L}{1-2\alpha}\left(\frac{1}{T^{1/3}}\sum_{t=s}^{T}\|\mathbf{v}_t\|^2\right)^{1-2\alpha}$$

$$=\frac{L}{1-2\alpha}\left(\frac{8-16\alpha}{1-\alpha}\right)^{\frac{1-2\alpha}{1-\alpha}}\left(\frac{1-\alpha}{8-16\alpha}\right)^{\frac{1-2\alpha}{1-\alpha}}\left(\frac{1}{T^{1/3}}\sum_{t=s}^{T}\|\mathbf{v}_t\|^2\right)^{1-2\alpha}$$

$$\leq\frac{\alpha}{1-\alpha}\left(\frac{L}{1-2\alpha}\left(\frac{8-16\alpha}{1-\alpha}\right)^{\frac{1-2\alpha}{1-\alpha}}\right)^{\frac{1-\alpha}{\alpha}}+\frac{1}{8}\left(\frac{1}{T^{1/3}}\sum_{t=s}^{T}\|\mathbf{v}_t\|^2\right)^{1-\alpha},$$

where the second inequality employs Young's inequality, such that $xy \leq \frac{\alpha}{1-\alpha}x^{\frac{1-\alpha}{\alpha}}+\frac{1-2\alpha}{1-\alpha}y^{\frac{1-\alpha}{1-2\alpha}}$ for positive $x, y$. Combining all the above, we know that

$$(\mathrm{B}) \leq L+2\alpha\left(\frac{(8-16\alpha)^{1-2\alpha}L}{1-2\alpha}\right)^{\frac{1}{2\alpha}}+\frac{\alpha}{1-\alpha}\left(\frac{L}{1-2\alpha}\left(\frac{8-16\alpha}{1-\alpha}\right)^{\frac{1-2\alpha}{1-\alpha}}\right)^{\frac{1-\alpha}{\alpha}}+\frac{\Gamma}{8}.$$

**Bounding** (C)**:** Given that $\beta = T^{-2/3}$ and $\eta_t \leq T^{-1/3}$, we can easily know that $(\mathrm{C}) \leq 2\sigma^2$.

So far, we bound all terms in Lemma 2, and we can deduce

$$\mathbf{LHS} \geq \mathbb{E}\left[\frac{1}{T^{1/3}}\sum_{t=1}^{s-1}\|\mathbf{v}_t\|^2\right]+\mathbb{E}\left[\frac{\Gamma}{2}\right];\quad \mathbf{RHS} \leq \mathbb{E}\left[\frac{\Gamma}{4}\right]+C_0,$$

where

$$C_0 = \left(2\Delta_f+3\sigma^2+L+2L^2\right)+3\alpha\left(\frac{2L^2}{1-3\alpha}(8-24\alpha)^{1-3\alpha}\right)^{\frac{1}{3\alpha}}$$

$$+\frac{2\alpha}{1-\alpha}\left(\frac{2L^2}{1-3\alpha}\left(\frac{8-24\alpha}{1-\alpha}\right)^{\frac{1-3\alpha}{1-\alpha}}\right)^{\frac{1-\alpha}{2\alpha}}+2\alpha\left(\frac{(8-16\alpha)^{1-2\alpha}L}{1-2\alpha}\right)^{\frac{1}{2\alpha}} \qquad (15)$$

$$+\frac{\alpha}{1-\alpha}\left(\frac{L}{1-2\alpha}\left(\frac{8-16\alpha}{1-\alpha}\right)^{\frac{1-2\alpha}{1-\alpha}}\right)^{\frac{1-\alpha}{\alpha}}.$$

These suggest that

$$\mathbb{E}\left[\sum_{t=1}^{s-1}\|\mathbf{v}_t\|^2\right] \leq C_0 T^{1/3};\quad \mathbb{E}\left[\Gamma\right] \leq 4C_0.$$

With the definition of $\Gamma$, we know that

$$\mathbb{E}\left[\min\left\{\frac{1}{T^{1/3}}\sum_{t=s}^{T}\|\mathbf{v}_t\|^2,\left(\frac{1}{T^{1/3}}\sum_{t=s}^{T}\|\mathbf{v}_t\|^2\right)^{1-\alpha}\right\}\right] \leq 4C_0,$$

which indicates the following by applying Jensen's inequality:

$$\mathbb{E}\left[\frac{1}{T}\sum_{t=s}^{T}\|\mathbf{v}_t\|\right] \leq \max\{(4C_0)^{\frac{1}{2}}, (4C_0)^{\frac{1}{2(1-\alpha)}}\}T^{-1/3}.$$

Also, because of Jensen's inequality, we have:

$$\mathbb{E}\left[\frac{1}{T}\sum_{t=1}^{s-1}\|\mathbf{v}_t\|\right] \leq \sqrt{\mathbb{E}\left[\left(\frac{1}{T}\sum_{t=1}^{s-1}\|\mathbf{v}_t\|\right)^2\right]} = \sqrt{\mathbb{E}\left[\frac{1}{T^2}\left(\sum_{t=1}^{s-1}\|\mathbf{v}_t\|\right)^2\right]}$$

$$\leq \sqrt{\mathbb{E}\left[\frac{s}{T^2}\sum_{t=1}^{s-1}\|\mathbf{v}_t\|^2\right]} \leq \sqrt{\mathbb{E}\left[\frac{1}{T}\sum_{t=1}^{s-1}\|\mathbf{v}_t\|^2\right]}$$

$$\leq \sqrt{\frac{1}{T}C_0 T^{1/3}} = \sqrt{C_0}T^{-1/3}.$$

Summing up, we have proven that

$$\mathbb{E}\left[\frac{1}{T}\sum_{t=1}^{T}\|\mathbf{v}_t\|\right] \leq \max\{3(C_0)^{\frac{1}{2}}, (C_0)^{\frac{1}{2}} + (4C_0)^{\frac{1}{2(1-\alpha)}}\}T^{-1/3}. \tag{16}$$

Finally, we finish our proof by introducing the following lemma.

**Lemma 3** *Suppose $0 < \beta < 1$, our method ensures that*

$$\sum_{t=1}^{T}\mathbb{E}\left[\|\mathbf{v}_t - \nabla f(\mathbf{x}_t)\|^2\right] \leq 3\sigma^2 T^{1/3} + \mathbb{E}\left[\frac{2L^2}{\beta}\sum_{t=1}^{T}\eta_t^2\|\mathbf{v}_t\|^2\right].$$

**Proof 3** *First note that we have already proven the following in equation (11).*

$$\mathbb{E}\left[\|\nabla f(\mathbf{x}_{t+1}) - \mathbf{v}_{t+1}\|^2\right] \leq (1-\beta)\mathbb{E}\left[\|\mathbf{v}_t - \nabla f(\mathbf{x}_t)\|^2\right] + 2\beta^2\sigma^2 + 2L^2\mathbb{E}\left[\eta_t^2\|\mathbf{v}_t\|^2\right].$$

*After rearranging the items, we can get the following:*

$$\mathbb{E}\left[\|\mathbf{v}_t - \nabla f(\mathbf{x}_t)\|^2\right] \leq \frac{1}{\beta}\left(\mathbb{E}\left[\|\mathbf{v}_t - \nabla f(\mathbf{x}_t)\|^2\right] - \mathbb{E}\left[\|\mathbf{v}_{t+1} - \nabla f(\mathbf{x}_{t+1})\|^2\right]\right) + 2\beta\sigma^2$$

$$+ \frac{2L^2}{\beta}\mathbb{E}\left[\eta_t^2\|\mathbf{v}_t\|^2\right].$$

*By summing up, we have:*

$$\sum_{t=1}^{T}\mathbb{E}\left[\|\mathbf{v}_t - \nabla f(\mathbf{x}_t)\|^2\right] \leq \frac{1}{\beta}\mathbb{E}\left[\|\mathbf{v}_1 - \nabla f(\mathbf{x}_1)\|^2\right] + 2\beta\sigma^2 T + \frac{2L^2}{\beta}\sum_{t=1}^{T}\mathbb{E}\left[\eta_t^2\|\mathbf{v}_t\|^2\right]. \tag{17}$$

*Since we use a large batch size in the first iteration, that is, $B_0 = T^{1/3}$, we can now ensure that $\mathbb{E}\left[\|\mathbf{v}_1 - \nabla f(\mathbf{x}_1)\|^2\right] \leq \frac{\sigma^2}{B_0} = \frac{\sigma^2}{T^{1/3}}$. Note that $\beta = \frac{1}{T^{2/3}}$, so first term equals to $\sigma^2 T^{1/3}$ and the second term reduces to $2\sigma^2 T^{1/3}$. To this end, we ensure*

$$\sum_{t=1}^{T}\mathbb{E}\left[\|\mathbf{v}_t - \nabla f(\mathbf{x}_t)\|^2\right] \leq 3\sigma^2 T^{1/3} + \frac{2L^2}{\beta}\sum_{t=1}^{T}\mathbb{E}\left[\eta_t^2\|\mathbf{v}_t\|^2\right].$$

*Thus we finish the proof for this lemma.*

Here, we bound the term $\frac{2L^2}{\beta}\sum_{t=1}^{T}\eta_t^2\|\mathbf{v}_t\|^2$ as follows. In the first stage, with $\eta_t = \frac{1}{T^{1/3}}$, we have:

$$\frac{2L^2}{\beta}\sum_{t=1}^{s-1}\eta_t^2\|\mathbf{v}_t\|^2 = \frac{2L^2}{\beta}\sum_{t=1}^{s-1}\frac{1}{T^{2/3}}\|\mathbf{v}_t\|^2 \leq 2L^2 T^{1/3}.$$

For the second stage, the analysis gives:

$$\frac{2L^2}{\beta}\sum_{t=s}^{T}\eta_t^2\,\|\mathbf{v}_t\|^2 \leq 2L^2 T^{2/3}\sum_{t=s}^{T}\frac{\|\mathbf{v}_t\|^2}{T^{\frac{2-2\alpha}{3}}(\sum_{i=s}^{t}\|\mathbf{v}_i\|)^{2\alpha}}$$

$$\leq \frac{2L^2 T^{2\alpha/3}}{1-2\alpha}\frac{((1-2\alpha)/L)^{1-2\alpha}}{((1-2\alpha)/L)^{1-2\alpha}}\left(\sum_{t=s}^{T}\|\mathbf{v}_t\|^2\right)^{1-2\alpha}$$

$$\leq 2\alpha\left(\frac{2L^2 T^{2\alpha/3}((1-2\alpha)/L)^{1-2\alpha}}{1-2\alpha}\right)^{1/2\alpha}+L\sum_{t=s}^{T}\|\mathbf{v}_t\|^2$$

$$\leq 2\alpha\left(\frac{2L^2((1-2\alpha)/L)^{1-2\alpha}}{1-2\alpha}\right)^{\frac{1}{2\alpha}}T^{1/3}+L\sum_{t=1}^{T}\|\mathbf{v}_t\|^2\,,$$

where the second inequality uses Lemma 1, and the third one employs Young's inequality, such that $xy\leq 2\alpha x^{\frac{1}{2\alpha}}+(1-2\alpha)y^{\frac{1}{1-2\alpha}}$ for positive $x,y$. We also know that

$$\frac{2L^2}{\beta}\sum_{t=s}^{T}\eta_t^2\,\|\mathbf{v}_t\|^2 \leq 2L^2 T^{2/3}\sum_{t=s}^{T}\frac{\|\mathbf{v}_t\|^2}{T^{\frac{2-2\alpha}{3}}(\sum_{i=s}^{t}\|\mathbf{v}_i\|)^{2\alpha}}$$

$$\leq \frac{2L^2 T^{2\alpha/3}}{1-2\alpha}\left(\frac{1-2\alpha}{(1-\alpha)L}T^{-\frac{\alpha}{3}}\right)^{\frac{1-2\alpha}{1-\alpha}}\left(\frac{1-\alpha}{1-2\alpha}T^{\frac{\alpha}{3}}L\right)^{\frac{1-2\alpha}{1-\alpha}}\left(\sum_{t=s}^{T}\|\mathbf{v}_t\|^2\right)^{1-2\alpha}$$

$$\leq \frac{\alpha}{1-\alpha}\left(\frac{2L^2 T^{2\alpha/3}}{1-2\alpha}\left(\frac{1-2\alpha}{(1-\alpha)L}T^{-\frac{\alpha}{3}}\right)^{\frac{1-2\alpha}{1-\alpha}}\right)^{\frac{1-\alpha}{\alpha}}+T^{\frac{\alpha}{3}}L\left(\sum_{t=s}^{T}\|\mathbf{v}_t\|^2\right)^{1-\alpha}$$

$$\leq \frac{\alpha}{1-\alpha}\left(\frac{2L^2}{1-2\alpha}\left(\frac{1-2\alpha}{(1-\alpha)L}\right)^{\frac{1-2\alpha}{1-\alpha}}\right)^{\frac{1-\alpha}{\alpha}}T^{1/3}+T^{\frac{\alpha}{3}}L\left(\sum_{t=s}^{T}\|\mathbf{v}_t\|^2\right)^{1-\alpha}\,,$$

where the second inequality uses Lemma 1, and the third one employs Young's inequality, such that $xy\leq\frac{\alpha}{1-\alpha}x^{\frac{1-\alpha}{\alpha}}+\frac{1-2\alpha}{1-\alpha}y^{\frac{1-\alpha}{1-2\alpha}}$ for positive $x,y$.

As a result, we have:

$$\sum_{t=1}^{T}\mathbb{E}\left[\|\mathbf{v}_t-\nabla f(\mathbf{x}_t)\|^2\right]$$

$$\leq\left(3\sigma^2+4C_0 L+L^{\frac{1+2\alpha}{2\alpha}}\left(\frac{2(1-2\alpha)^{1-2\alpha}}{1-2\alpha}\right)^{\frac{1}{2\alpha}}+L^{\frac{1}{\alpha}}\left(\frac{2}{1-2\alpha}\left(\frac{1-2\alpha}{1-\alpha}\right)^{\frac{1-2\alpha}{1-\alpha}}\right)^{\frac{1-\alpha}{\alpha}}\right)T^{1/3}.$$

By integrating these findings, we can finally have:

$$\mathbb{E}\left[\frac{1}{T}\sum_{t=1}^{T}\|\nabla f(\mathbf{x}_t)\|\right]\leq\frac{1}{T}\mathbb{E}\left[\sum_{t=1}^{T}\|\mathbf{v}_t\|\right]+\frac{1}{T}\left[\sum_{t=1}^{T}\|\nabla f(\mathbf{x}_t)-\mathbf{v}_t\|\right]\leq\frac{C'}{T^{1/3}},$$

where

$$C'=\max\{3(C_0)^{\frac{1}{2}},(C_0)^{\frac{1}{2}}+(4C_0)^{\frac{1}{2(1-\alpha)}}\}$$

$$+\sqrt{3\sigma^2+4C_0 L+L^{\frac{1+2\alpha}{2\alpha}}\left(\frac{2(1-2\alpha)^{1-2\alpha}}{1-2\alpha}\right)^{\frac{1}{2\alpha}}+L^{\frac{1}{\alpha}}\left(\frac{2}{1-2\alpha}\left(\frac{1-2\alpha}{1-\alpha}\right)^{\frac{1-2\alpha}{1-\alpha}}\right)^{\frac{1-\alpha}{\alpha}}}$$

$$=\mathcal{O}\left(\Delta_f^{\frac{1}{2(1-\alpha)}}+\sigma^{\frac{1}{1-\alpha}}+L^{\frac{1}{2\alpha}}\right),$$

with $C_0$ defined in equation (15). We find that larger $\alpha$ leads to better dependence on $L$ and worse reliance on parameters $\Delta$ and $\sigma$. For $\alpha \to \frac{1}{3}$, we can obtain that

$$\mathbb{E}\left[\frac{1}{T}\sum_{t=1}^{T}\|\nabla f(\mathbf{x}_t)\|\right] \leq \mathcal{O}\left(\frac{\Delta_f^{3/4} + \sigma^{3/2} + L^{3/2}}{T^{1/3}}\right).$$

Since we require $0 < \alpha < \frac{1}{3}$, in practice, we can use $\alpha = 0.3$ instead, which leads to a convergence rate of $\mathcal{O}\left(\frac{\Delta_f^{5/7} + \sigma^{10/7} + L^{5/3}}{T^{1/3}}\right)$.

## B   Proof of Theorem 2

Since $2^0 + 2^1 + \cdots + 2^{K-1} < 2^K$, running the algorithm for $T$ iterations guarantees at least $K = \lfloor \log(T) \rfloor$ complete stages. In the theoretical analysis, we can simply use the output of the last complete stage $K = \lfloor \log(T) \rfloor$, which has been at least run for $2^{K-1} \geq T/4$ iterations. According to the analysis of Theorem 1, we have already known that running the Algorithm 1 for $T/4$ iterations leads to the following guarantee:

$$\mathbb{E}\left[\|\nabla f(\mathbf{x}_\tau)\|\right] \leq \mathcal{O}\left(\frac{\Delta_f^{\frac{1}{2(1-\alpha)}} + \sigma^{\frac{1}{1-\alpha}} + L^{\frac{1}{2\alpha}}}{(T/4)^{1/3}}\right) = \mathcal{O}\left(\frac{\Delta_f^{\frac{1}{2(1-\alpha)}} + \sigma^{\frac{1}{1-\alpha}} + L^{\frac{1}{2\alpha}}}{T^{1/3}}\right),$$

which is on the same order of the original convergence rate.

## C   Proof of Theorem 3

According to equation (14), we have already proven that

$$\sum_{t=1}^{T}\eta_t\|\mathbf{v}_t\|^2 \leq 2F(\mathbf{x}_1) - 2F(\mathbf{x}_{T+1}) + \sum_{t=1}^{T}\eta_t\|\nabla F(\mathbf{x}_t) - \mathbf{v}_t\|^2 + \sum_{t=1}^{T}\eta_t^2 L\|\mathbf{v}_t\|^2.$$

Then we bound the term $\sum_{t=1}^{T}\eta_t\|\nabla F(\mathbf{x}_t) - \mathbf{v}_t\|^2$ as follows:

$$\|\nabla F(\mathbf{x}_t) - \mathbf{v}_t\|^2 \leq 2\|\nabla f(g(\mathbf{x}_t))\nabla g(\mathbf{x}_t) - \nabla f(\mathbf{u}_t)\nabla g(\mathbf{x}_t)\|^2 + 2\|\nabla f(\mathbf{u}_t)\nabla g(\mathbf{x}_t) - \mathbf{v}_t\|^2$$
$$\leq 2C^2L^2\|g(\mathbf{x}_t) - \mathbf{u}_t\|^2 + 2\|\mathbf{v}_t - \nabla f(\mathbf{u}_t)\nabla g(\mathbf{x}_t)\|^2.$$

Define that $G_t = \nabla f(\mathbf{u}_t)\nabla g(\mathbf{x}_t)$, then we have:

$$\sum_{t=1}^{T}\mathbb{E}\left[\eta_t\|\nabla F(\mathbf{x}_t) - \mathbf{v}_t\|^2\right] \leq 2C^2L^2\sum_{t=1}^{T}\mathbb{E}\left[\eta_t\|g(\mathbf{x}_t) - \mathbf{u}_t\|^2\right] + 2\sum_{t=1}^{T}\mathbb{E}\left[\eta_t\|\mathbf{v}_t - G_t\|^2\right].$$

For the term $\sum_{t=1}^{T}\mathbb{E}\left[\eta_t\|\mathbf{v}_t - G_t\|^2\right]$, following the very similar analysis of equation (11), we have the following guarantee:

$$\mathbb{E}_{\xi_{t+1},\zeta_{t+1}}\left[\|\mathbf{v}_{t+1} - G_{t+1}\|^2\right]$$
$$\leq (1-\beta)\|\mathbf{v}_t - G_t\|^2$$
$$\quad + 2\beta^2\mathbb{E}_{\xi_{t+1},\zeta_{t+1}}\left[\|\nabla f(\mathbf{u}_{t+1};\xi_{t+1})\nabla g(\mathbf{x}_{t+1};\zeta_{t+1}) - \nabla f(\mathbf{u}_{t+1})\nabla g(\mathbf{x}_{t+1})\|^2\right]$$
$$\quad + 2\mathbb{E}_{\xi_{t+1},\zeta_{t+1}}\left[\|\nabla f(\mathbf{u}_{t+1};\xi_{t+1})\nabla g(\mathbf{x}_{t+1};\zeta_{t+1}) - \nabla f(\mathbf{u}_t;\xi_{t+1})\nabla g(\mathbf{x}_t;\zeta_{t+1})\|^2\right]$$
$$\leq (1-\beta)\|\mathbf{v}_t - G_t\|^2 + 4C^2\sigma^2\beta^2 + 4C^2L^2\mathbb{E}\left[\eta_t^2\|\mathbf{v}_t\|^2\right] + 4C^2L^2\mathbb{E}\left[\|\mathbf{u}_{t+1} - \mathbf{u}_t\|^2\right].$$

That is to say:

$$\sum_{t=1}^{T} \mathbb{E}\left[\eta_t \left\|\mathbf{v}_t - G_t\right\|^2\right]$$

$$\leq \mathbb{E}\left[\frac{\eta_1}{\beta} \left\|\mathbf{v}_1 - G_1\right\|^2\right] + 4C^2\sigma^2\beta\mathbb{E}\left[\sum_{t=1}^{T}\eta_t\right] + \frac{4C^2L^2}{\beta}\sum_{t=1}^{T}\mathbb{E}\left[\eta_t^3 \left\|\mathbf{v}_t\right\|^2\right]$$

$$+ \frac{4C^2L^2}{\beta}\sum_{t=1}^{T}\mathbb{E}\left[\eta_t \left\|\mathbf{u}_{t+1} - \mathbf{u}_t\right\|^2\right]$$

$$\leq (1 + 4C^2)\sigma^2 + \frac{4C^2L^2}{\beta}\sum_{t=1}^{T}\mathbb{E}\left[\eta_t^3 \left\|\mathbf{v}_t\right\|^2\right] + \frac{4C^2L^2}{\beta}\sum_{t=1}^{T}\mathbb{E}\left[\eta_t \left\|\mathbf{u}_{t+1} - \mathbf{u}_t\right\|^2\right],$$

where the last inequality due to the fact that $\beta = T^{-2/3}$, $\eta_t \leq T^{-1/3}$, and we use a large batch size $T^{1/3}$ in the first iteration. Next, we further bound the term $\sum_{t=1}^{T} \mathbb{E}\left[\eta_{t-1} \left\|\mathbf{u}_t - \mathbf{u}_{t-1}\right\|^2\right]$. First, we can ensure that:

$$\mathbb{E}_{\zeta_t}\left[\left\|\mathbf{u}_t - \mathbf{u}_{t-1}\right\|^2\right]$$

$$= \mathbb{E}_{\zeta_t}\left[\left\|\beta(g(\mathbf{x}_t; \zeta_t) - \mathbf{u}_{t-1}) + (1 - \beta)(g(\mathbf{x}_t; \zeta_t) - g(\mathbf{x}_{t-1}; \zeta_t))\right\|^2\right]$$

$$= \mathbb{E}_{\zeta_t}\left[\left\|\beta(g(\mathbf{x}_{t-1}) - \mathbf{u}_{t-1}) + (g(\mathbf{x}_t; \zeta_t) - g(\mathbf{x}_{t-1}; \zeta_t)) + \beta(g(\mathbf{x}_{t-1}; \zeta_t) - g(\mathbf{x}_{t-1}))\right\|^2\right]$$

$$\leq 3\beta^2 \left\|g(\mathbf{x}_{t-1}) - \mathbf{u}_{t-1}\right\|^2 + 3C^2\eta_{t-1}^2 \left\|\mathbf{v}_{t-1}\right\|^2 + 3\beta^2\sigma^2.$$

So we know that

$$\frac{1}{\beta}\sum_{t=1}^{T}\mathbb{E}\left[\eta_t \left\|\mathbf{u}_{t+1} - \mathbf{u}_t\right\|^2\right]$$

$$\leq 3\beta\sum_{t=1}^{T}\mathbb{E}\left[\eta_t \left\|g(\mathbf{x}_t) - \mathbf{u}_t\right\|^2\right] + 3C^2\mathbb{E}\left[\sum_{t=1}^{T}\frac{\eta_t^3}{\beta}\left\|\mathbf{v}_t\right\|^2\right] + 3\beta\sigma^2\mathbb{E}\left[\sum_{t=1}^{T}\eta_t\right].$$

So far, we have

$$\sum_{t=1}^{T}\mathbb{E}\left[\eta_t \left\|\nabla F(\mathbf{x}_t) - \mathbf{v}_t\right\|^2\right] \leq 26C^2L^2\sum_{t=1}^{T}\mathbb{E}\left[\eta_t \left\|\mathbf{u}_t - g(\mathbf{x}_t)\right\|^2\right]$$

$$+ \frac{8C^2L^2 + 24C^4L^2}{\beta}\sum_{t=1}^{T}\mathbb{E}\left[\eta_t^3 \left\|\mathbf{v}_t\right\|^2\right] + (2 + 8C^2 + 24C^2L^2)\sigma^2.$$

Next, we can bound $\sum_{t=1}^{T} \eta_t \left\|\mathbf{u}_t - g(\mathbf{x}_t)\right\|^2$ following equation (11), as:

$$\left\|\mathbf{u}_t - g(\mathbf{x}_t)\right\|^2 \leq \frac{1}{\beta}\left(\left\|\mathbf{u}_t - g(\mathbf{x}_t)\right\|^2 - \mathbb{E}_{\zeta_{t+1}}\left[\left\|\mathbf{u}_{t+1} - g(\mathbf{x}_{t+1})\right\|^2\right]\right) + 2\beta\sigma^2 + \frac{2C^2\eta_t^2}{\beta}\left\|\mathbf{v}_t\right\|^2.$$

So we can have

$$\mathbb{E}\left[\sum_{t=1}^{T}\eta_t \left\|\mathbf{u}_t - g(\mathbf{x}_t)\right\|^2\right] \leq \mathbb{E}\left[\frac{\eta_1}{\beta}\left\|\mathbf{u}_1 - g(\mathbf{x}_1)\right\|^2\right] + 2\beta\sigma^2\mathbb{E}\left[\sum_{t=1}^{T}\eta_t\right] + \frac{2C^2}{\beta}\sum_{t=1}^{T}\mathbb{E}\left[\eta_t^3 \left\|\mathbf{v}_t\right\|^2\right]$$

$$\leq 3\sigma^2 + \frac{2C^2}{\beta}\sum_{t=1}^{T}\mathbb{E}\left[\eta_t^3 \left\|\mathbf{v}_t\right\|^2\right],$$

where the last inequality due to the fact that $\beta = T^{-2/3}$, $\eta_t \leq T^{-1/3}$, and we use a large batch size $T^{1/3}$ in the first iteration. Combining all, we have:

$$\sum_{t=1}^{T} \mathbb{E}\left[\eta_t \|\mathbf{v}_t\|^2\right] \leq 2\Delta_F + (2 + 8C^2 + 102C^2L^2)\sigma^2$$

$$+ (8C^2L^2 + 76C^4L^2)\mathbb{E}\left[\sum_{t=1}^{T} \frac{\eta_t^3}{\beta} \|\mathbf{v}_t\|^2\right] + L\mathbb{E}\left[\sum_{t=1}^{T} \eta_t^2 \|\mathbf{v}_t\|^2\right].$$

Treating $\Delta_F, C, L, \sigma$ as constant, the above inequality is very similar to Lemma 2. Thus following the very similar analysis after Lemma 2, we can show that:

$$\mathbb{E}\left[\frac{1}{T}\sum_{t=1}^{T} \|\mathbf{v}_t\|\right] \leq \mathcal{O}(T^{-1/3}).$$

According to previous analysis, we also have that

$$\sum_{t=1}^{T} \mathbb{E}\left[\|\nabla f(\mathbf{x}_t) - \mathbf{v}_t\|^2\right]$$

$$\leq 26C^2L^2 \sum_{t=1}^{T} \mathbb{E}\left[\|\mathbf{u}_t - g(\mathbf{x}_t)\|^2\right] + \frac{8C^2L^2 + 24C^4L^2}{\beta}\mathbb{E}\left[\sum_{t=1}^{T} \eta_t^2 \|\mathbf{v}_t\|^2\right]$$

$$+ (2 + 8C^2 + 24C^2L^2)\sigma^2 T^{1/3}$$

$$\leq (2 + 8C^2 + 102C^2L^2)\sigma^2 T^{1/3} + 8C^2L^2 + 76C^4L^2\mathbb{E}\left[\sum_{t=1}^{T} \frac{\eta_t^2}{\beta} \|\mathbf{v}_t\|^2\right],$$

which is similar to Lemma 3, and leads to $\sum_{t=1}^{T} \mathbb{E}\left[\|\nabla f(\mathbf{x}_t) - \mathbf{v}_t\| / T\right] \leq \mathcal{O}(T^{-1/3})$ following the same analysis. Combing all these together, we can deduce that

$$\frac{1}{T}\sum_{t=1}^{T} \|\nabla F(\mathbf{x}_t)\| \leq \mathcal{O}(T^{-1/3}),$$

which finishes the proof of Theorem 3.

## D    Proof of Theorem 4

Due to the smoothness of $F(\mathbf{x})$, we have proven the following in equation (14):

$$\sum_{t=1}^{T} \eta_t \|\mathbf{v}_t\|^2 \leq 2F(\mathbf{x}_1) - 2F(\mathbf{x}_{T+1}) + \sum_{t=1}^{T} \eta_t \|\nabla F(\mathbf{x}_t) - \mathbf{v}_t\|^2 + \sum_{t=1}^{T} \eta_t^2 L \|\mathbf{v}_t\|^2.$$

For the LHS, we have the following guarantee:

$$\sum_{t=1}^{T} \eta_t \|\mathbf{v}_t\|^2 = \sum_{t=1}^{T} \frac{\|\mathbf{v}_t\|^2}{n^{\frac{1-\alpha}{2}}\left(\sum_{i=1}^{t} \|\mathbf{v}_i\|^2\right)^{\alpha}} \geq \left(\frac{1}{\sqrt{n}}\sum_{t=1}^{T} \|\mathbf{v}_t\|^2\right)^{1-\alpha}.$$

Then, we bound the terms in the RHS. First, we have the following lemma.

**Lemma 4** *Define that $\mathbf{z}_t = \nabla f_{i_t}(\mathbf{x}_t) - g_t^{i_t} + \frac{1}{n}\sum_{i=1}^{n} g_t^i$, we have:*

$$\mathbb{E}\left[\sum_{t=1}^{T} \eta_t \|\nabla F(\mathbf{x}_t) - \mathbf{v}_t\|^2\right] \leq 2\beta\mathbb{E}\left[\sum_{t=1}^{T} \eta_t \|\nabla F(\mathbf{x}_{t+1}) - \mathbf{z}_{t+1}\|^2\right] + 2L^2\mathbb{E}\left[\sum_{t=1}^{T} \frac{\eta_t^3}{\beta} \|\mathbf{v}_t\|^2\right].$$

**Proof 4** *According to the definition of $\mathbf{z}_t$, the estimator $\mathbf{v}_t$ can be expressed as*

$$\mathbf{v}_t = (1-\beta)\mathbf{v}_{t-1} + \beta\mathbf{z}_t + (1-\beta)\left(\nabla f_{i_t}(\mathbf{x}_t) - \nabla f_{i_t}(\mathbf{x}_{t-1})\right).$$

*Note that $\mathbb{E}_{i_{t+1}}\left[\mathbf{z}_{t+1}\right] = \nabla F(\mathbf{x}_{t+1})$, and we have:*

$$
\begin{aligned}
&\mathbb{E}_{i_{t+1}}\left[\|\nabla F(\mathbf{x}_{t+1}) - \mathbf{v}_{t+1}\|^2\right] \\
&= \mathbb{E}_{i_{t+1}}\left[\left\|(1-\beta)\mathbf{v}_t + \beta\mathbf{z}_{t+1} + (1-\beta)\left(\nabla f_{i_{t+1}}(\mathbf{x}_{t+1}) - \nabla f_{i_{t+1}}(\mathbf{x}_t)\right) - \nabla F(\mathbf{x}_{t+1})\right\|^2\right] \\
&= \mathbb{E}_{i_{t+1}}\left[\|(1-\beta)(\mathbf{v}_t - \nabla F(\mathbf{x}_t)) + \beta(\mathbf{z}_{t+1} - \nabla F(\mathbf{x}_{t+1}))\right. \\
&\qquad\left. +(1-\beta)\left(\nabla f_{i_{t+1}}(\mathbf{x}_{t+1}) - \nabla f_{i_{t+1}}(\mathbf{x}_t) + \nabla F(\mathbf{x}_t) - \nabla F(\mathbf{x}_{t+1})\right)\|^2\right] \\
&\leq \|(1-\beta)(\mathbf{v}_t - \nabla F(\mathbf{x}_t))\|^2 + \mathbb{E}_{i_{t+1}}\left[\|\beta(\mathbf{z}_{t+1} - \nabla F(\mathbf{x}_{t+1}))\right. \\
&\qquad\left. +(1-\beta)\left(\nabla f_{i_{t+1}}(\mathbf{x}_{t+1}) - \nabla f_{i_{t+1}}(\mathbf{x}_t) + \nabla F(\mathbf{x}_t) - \nabla F(\mathbf{x}_{t+1})\right)\|^2\right] \\
&\leq (1-\beta)^2\|\mathbf{v}_t - \nabla F(\mathbf{x}_t)\|^2 + 2\beta^2\mathbb{E}_{i_{t+1}}\left[\|\mathbf{z}_{t+1} - \nabla F(\mathbf{x}_{t+1})\|^2\right] \\
&\qquad + 2(1-\beta)^2\mathbb{E}_{i_{t+1}}\left[\left\|\nabla f_{i_{t+1}}(\mathbf{x}_{t+1}) - \nabla f_{i_{t+1}}(\mathbf{x}_t) + \nabla F(\mathbf{x}_t) - \nabla F(\mathbf{x}_{t+1})\right\|^2\right] \\
&\leq (1-\beta)^2\|\mathbf{v}_t - \nabla F(\mathbf{x}_t)\|^2 + 2\beta^2\mathbb{E}_{i_{t+1}}\left[\|\mathbf{z}_{t+1} - \nabla F(\mathbf{x}_{t+1})\|^2\right] \\
&\qquad + 2(1-\beta)^2\mathbb{E}_{i_{t+1}}\left[\left\|\nabla f_{i_{t+1}}(\mathbf{x}_{t+1}) - \nabla f_{i_{t+1}}(\mathbf{x}_t)\right\|^2\right] \\
&\leq (1-\beta)^2\|\mathbf{v}_t - \nabla F(\mathbf{x}_t)\|^2 + 2\beta^2\mathbb{E}_{i_{t+1}}\left[\|\mathbf{z}_{t+1} - \nabla F(\mathbf{x}_{t+1})\|^2\right] + 2L^2\|\mathbf{x}_{t+1} - \mathbf{x}_t\|^2 \\
&\leq (1-\beta)\|\mathbf{v}_t - \nabla F(\mathbf{x}_t)\|^2 + 2\beta^2\mathbb{E}_{i_{t+1}}\left[\|\mathbf{z}_{t+1} - \nabla F(\mathbf{x}_{t+1})\|^2\right] + 2L^2\eta_t^2\|\mathbf{v}_t\|^2.
\end{aligned}
\tag{18}
$$

*Rearrange the items and multiply the both sides by $\eta_t$, we can get the following:*

$$
\begin{aligned}
\mathbb{E}\left[\eta_t\|\mathbf{v}_t - \nabla F(\mathbf{x}_t)\|^2\right] \leq & \mathbb{E}\left[\frac{\eta_t}{\beta}\|\mathbf{v}_t - \nabla F(\mathbf{x}_t)\|^2\right] - \mathbb{E}\left[\frac{\eta_t}{\beta}\|\mathbf{v}_{t+1} - \nabla F(\mathbf{x}_{t+1})\|^2\right] \\
& + 2\beta\mathbb{E}\left[\eta_t\|\mathbf{z}_{t+1} - \nabla F(\mathbf{x}_{t+1})\|^2\right] + \frac{2L^2}{\beta}\mathbb{E}\left[\eta_t^3\|\mathbf{v}_t\|^2\right].
\end{aligned}
$$

*Note that $\eta_t$ is non-increasing. By summing up, we have:*

$$
\begin{aligned}
&\sum_{t=1}^{T}\mathbb{E}\left[\eta_t\|\mathbf{v}_t - \nabla F(\mathbf{x}_t)\|^2\right] \\
&\leq \mathbb{E}\left[\frac{\eta_1}{\beta}\|\mathbf{v}_1 - \nabla F(\mathbf{x}_1)\|^2\right] + 2\beta\sum_{t=1}^{T}\mathbb{E}\left[\eta_t\|\mathbf{z}_{t+1} - \nabla F(\mathbf{x}_{t+1})\|^2\right] + \frac{2L^2}{\beta}\sum_{t=1}^{T}\mathbb{E}\left[\eta_t^3\|\mathbf{v}_t\|^2\right].
\end{aligned}
$$

*Since we use a full batch in the first iteration, we can finish the proof of this lemma.*

Next, we bound two terms in the above lemma.

**Lemma 5** *We can ensure that*

$$2\beta\sum_{t=1}^{T}\mathbb{E}\left[\eta_t\|\nabla F(\mathbf{x}_{t+1}) - \mathbf{z}_{t+1}\|^2\right] \leq 12L^2\sum_{t=1}^{T}\mathbb{E}\left[\frac{\eta_t^3}{\beta}\|\mathbf{v}_t\|^2\right].$$

**Proof 5**

$$\mathbb{E}_{i_{t+1}}\left[\eta_t \left\|\nabla F(\mathbf{x}_{t+1}) - \mathbf{z}_{t+1}\right\|^2\right]$$

$$=\mathbb{E}_{i_{t+1}}\left[\eta_t \left\|\nabla F(\mathbf{x}_{t+1}) - \nabla f_{i_{t+1}}(\mathbf{x}_{t+1}) + g_{t+1}^{i_{t+1}} - \frac{1}{n}\sum_{i=1}^{n} g_{t+1}^i\right\|^2\right]$$

$$=\mathbb{E}_{i_{t+1}}\left[\eta_t \left\|\nabla f_{i_{t+1}}(\mathbf{x}_{t+1}) - g_{t+1}^{i_{t+1}} - \left(\nabla F(\mathbf{x}_{t+1}) - \frac{1}{n}\sum_{i=1}^{n} g_{t+1}^i\right)\right\|^2\right] \quad (19)$$

$$\leq\mathbb{E}_{i_{t+1}}\left[\eta_t \left\|\nabla f_{i_{t+1}}(\mathbf{x}_{t+1}) - g_{t+1}^{i_{t+1}}\right\|^2\right]$$

$$=\frac{1}{n}\sum_{i=1}^{n} \eta_t \left\|\nabla f_i(\mathbf{x}_{t+1}) - g_{t+1}^i\right\|^2,$$

*where the last equation is due to the fact that $i_{t+1}$ is randomly sample from $\{1, 2, \cdots, n\}$. Note that we also have that*

$$\frac{1}{n}\sum_{i=1}^{n} \eta_t \left\|\nabla f_i(\mathbf{x}_{t+1}) - g_{t+1}^i\right\|^2 = \mathbb{E}_{i_{t+1}}\left[\eta_t \left\|\nabla f_{i_{t+1}}(\mathbf{x}_{t+1}) - g_{t+1}^{i_{t+1}}\right\|^2\right]$$

$$\leq\mathbb{E}_{i_{t+1}}\left[\eta_t(1 + 2n)\left\|\nabla f_{i_{t+1}}(\mathbf{x}_{t+1}) - \nabla f_{i_{t+1}}(\mathbf{x}_t)\right\|^2 + \eta_t(1 + \frac{1}{2n})\left\|\nabla f_{i_{t+1}}(\mathbf{x}_t) - g_{t+1}^{i_{t+1}}\right\|^2\right]$$

$$\leq\mathbb{E}_{i_{t+1}}\left[(1 + 2n)L^2\eta_t^3 \left\|\mathbf{v}_t\right\|^2 + \eta_t(1 + \frac{1}{2n})\left\|\nabla f_{i_{t+1}}(\mathbf{x}_t) - g_{t+1}^{i_{t+1}}\right\|^2\right]$$

$$\leq\mathbb{E}_{i_{t+1}}\left[(1 + 2n)L^2\eta_t^3 \left\|\mathbf{v}_t\right\|^2\right.$$

$$\left. +\eta_t(1 + \frac{1}{2n})\left((1 - \frac{1}{n})\left\|\nabla f_{i_{t+1}}(\mathbf{x}_t) - g_t^{i_{t+1}}\right\|^2 + \frac{1}{n}\left\|\nabla f_{i_{t+1}}(\mathbf{x}_t) - \nabla f_{i_{t+1}}(\mathbf{x}_t)\right\|^2\right)\right]$$

$$\leq\mathbb{E}_{i_{t+1}}\left[3nL^2\eta_t^3 \left\|\mathbf{v}_t\right\|^2 + \eta_t(1 - \frac{1}{2n})\left\|\nabla f_{i_{t+1}}(\mathbf{x}_t) - g_t^{i_{t+1}}\right\|^2\right]$$

$$\leq 3nL^2\eta_t^3 \left\|\mathbf{v}_t\right\|^2 + \left(1 - \frac{1}{2n}\right)\frac{1}{n}\sum_{i=1}^{n} \eta_t \left\|\nabla f_i(\mathbf{x}_t) - g_t^i\right\|^2.$$

*That is to say, we can ensure that*

$$\frac{1}{n}\sum_{i=1}^{n} \eta_{t+1} \left\|\nabla f_i(\mathbf{x}_{t+1}) - g_{t+1}^i\right\|^2 \leq \sum_{i=1}^{n} \eta_t \left\|\nabla f_i(\mathbf{x}_{t+1}) - g_{t+1}^i\right\|^2$$

$$\leq 3nL^2\eta_t^3 \left\|\mathbf{v}_t\right\|^2 + \left(1 - \frac{1}{2n}\right)\frac{1}{n}\sum_{i=1}^{n} \eta_t \left\|\nabla f_i(\mathbf{x}_t) - g_t^i\right\|^2.$$

*By rearranging and summing up, we have*

$$\frac{1}{2n}\sum_{t=1}^{T}\frac{1}{n}\sum_{i=1}^{n} \eta_t \left\|\nabla f_i(\mathbf{x}_t) - g_t^i\right\|^2 \leq 3nL^2\sum_{t=1}^{T}\eta_t^3 \left\|\mathbf{v}_t\right\|^2 + \frac{1}{n}\sum_{i=1}^{n} \eta_1 \left\|\nabla f_i(\mathbf{x}_1) - g_1^i\right\|^2.$$

*Since we use a full batch $n$ in the first iteration, the second term equals zero, and thus we obtain:*

$$\sum_{t=1}^{T}\frac{1}{n}\sum_{i=1}^{n} \eta_t \left\|\nabla f_i(\mathbf{x}_t) - g_t^i\right\|^2 \leq 6n^2L^2\sum_{t=1}^{T}\eta_t^3 \left\|\mathbf{v}_t\right\|^2 = \frac{6L^2}{\beta^2}\sum_{t=1}^{T}\eta_t^3 \left\|\mathbf{v}_t\right\|^2,$$

*which leads to the result of this lemma.*

**Lemma 6** *We have the following guarantee:*

$$2L^2\sum_{t=1}^{T}\frac{\eta_t^3}{\beta}\left\|\mathbf{v}_t\right\|^2 \leq \frac{2\alpha}{1-\alpha}\left(\frac{2L^2}{1-3\alpha}\left(\frac{24-72\alpha}{1-\alpha}\right)^{\frac{1-3\alpha}{1-\alpha}}\right)^{\frac{1-\alpha}{2\alpha}} + \frac{1}{24}\left(\frac{1}{\sqrt{n}}\sum_{t=1}^{T}\left\|\mathbf{v}_t\right\|^2\right)^{1-\alpha}.$$

**Proof 6**

$$2L^2 \sum_{t=1}^{T} \frac{\eta_t^3}{\beta} \|\mathbf{v}_t\|^2$$

$$=2L^2 n \sum_{t=1}^{T} \eta_t^3 \|\mathbf{v}_t\|^2 = 2L^2 \sum_{t=1}^{T} \frac{\|\mathbf{v}_t\|^2}{n^{\frac{1-3\alpha}{2}} \left(\sum_{i=1}^{t} \|\mathbf{v}_i\|^2\right)^{3\alpha}}$$

$$\leq \frac{2L^2}{1-3\alpha} \left(\frac{1}{\sqrt{n}} \sum_{t=1}^{T} \|\mathbf{v}_t\|^2\right)^{1-3\alpha}$$

$$=\frac{2L^2}{1-3\alpha} \left(\frac{24-72\alpha}{1-\alpha}\right)^{\frac{1-3\alpha}{1-\alpha}} \left(\frac{1-\alpha}{24-72\alpha}\right)^{\frac{1-3\alpha}{1-\alpha}} \left(\frac{1}{\sqrt{n}} \sum_{t=1}^{T} \|\mathbf{v}_t\|^2\right)^{1-3\alpha}$$

$$\leq \frac{2\alpha}{1-\alpha} \left(\frac{2L^2}{1-3\alpha} \left(\frac{24-72\alpha}{1-\alpha}\right)^{\frac{1-3\alpha}{1-\alpha}}\right)^{\frac{1-\alpha}{2\alpha}} + \frac{1}{24} \left(\frac{1}{\sqrt{n}} \sum_{t=1}^{T} \|\mathbf{v}_t\|^2\right)^{1-\alpha},$$

where the last inequality employs Young's inequality, such that $xy \leq \frac{2\alpha}{1-\alpha} x^{\frac{1-\alpha}{2\alpha}} + \frac{1-3\alpha}{1-\alpha} y^{\frac{1-\alpha}{1-3\alpha}}$ for positive $x, y$.

**Lemma 7** *We can ensure the following guarantee:*

$$\mathbb{E}\left[\sum_{t=1}^{T} \eta_t^2 L \|\mathbf{v}_t\|^2\right] \leq \frac{\alpha}{1-\alpha} \left(\frac{L}{1-2\alpha} \left(\frac{4-8\alpha}{1-\alpha}\right)^{\frac{1-2\alpha}{1-\alpha}}\right)^{\frac{1-\alpha}{\alpha}} + \frac{1}{4}\mathbb{E}\left[\left(\frac{1}{\sqrt{n}} \sum_{t=1}^{T} \|\mathbf{v}_t\|^2\right)^{1-\alpha}\right]$$

**Proof 7**

$$\sum_{t=1}^{T} \eta_t^2 L \|\mathbf{v}_t\|^2 = L \sum_{t=1}^{T} \frac{\|\mathbf{v}_t\|^2}{n^{1-\alpha} \left(\sum_{i=1}^{t} \|\mathbf{v}_i\|^2\right)^{2\alpha}}$$

$$\leq \frac{L}{1-2\alpha} \frac{1}{n^{1-\alpha}} \left(\sum_{t=1}^{T} \|\mathbf{v}_t\|^2\right)^{1-2\alpha}$$

$$\leq \frac{L}{1-2\alpha} \left(\frac{1}{\sqrt{n}} \sum_{t=1}^{T} \|\mathbf{v}_t\|^2\right)^{1-2\alpha}$$

$$=\frac{L}{1-2\alpha} \left(\frac{4-8\alpha}{1-\alpha}\right)^{\frac{1-2\alpha}{1-\alpha}} \left(\frac{1-\alpha}{4-8\alpha}\right)^{\frac{1-2\alpha}{1-\alpha}} \left(\frac{1}{\sqrt{n}} \sum_{t=1}^{T} \|\mathbf{v}_t\|^2\right)^{1-2\alpha}$$

$$\leq \frac{\alpha}{1-\alpha} \left(\frac{L}{1-2\alpha} \left(\frac{4-8\alpha}{1-\alpha}\right)^{\frac{1-2\alpha}{1-\alpha}}\right)^{\frac{1-\alpha}{\alpha}} + \frac{1}{4} \left(\frac{1}{\sqrt{n}} \sum_{t=1}^{T} \|\mathbf{v}_t\|^2\right)^{1-\alpha},$$

where the last inequality employs Young's inequality, such that $xy \leq \frac{\alpha}{1-\alpha} x^{\frac{1-\alpha}{\alpha}} + \frac{1-2\alpha}{1-\alpha} y^{\frac{1-\alpha}{1-2\alpha}}$ for positive $x, y$. Combing all these, we have already proven that

$$\mathbb{E}\left[\sum_{t=1}^{T} \eta_t \|\mathbf{v}_t\|^2\right] \geq \mathbb{E}\left[\left(\frac{1}{\sqrt{n}} \sum_{t=1}^{T} \|\mathbf{v}_t\|^2\right)^{1-\alpha}\right],$$

$$\sum_{t=1}^{T} \eta_t \|\mathbf{v}_t\|^2 \leq C_3 + \frac{3}{4}\mathbb{E}\left[\left(\frac{1}{\sqrt{n}} \sum_{t=1}^{T} \|\mathbf{v}_t\|^2\right)^{1-\alpha}\right],$$

where

$$C_3 = 2\Delta_F + \frac{14\alpha}{1-\alpha}\left(\frac{2L^2}{1-3\alpha}\left(\frac{24-72\alpha}{1-\alpha}\right)^{\frac{1-3\alpha}{1-\alpha}}\right)^{\frac{1-\alpha}{2\alpha}}$$

$$+ \frac{\alpha}{1-\alpha}\left(\frac{L}{1-2\alpha}\left(\frac{4-8\alpha}{1-\alpha}\right)^{\frac{1-2\alpha}{1-\alpha}}\right)^{\frac{1-\alpha}{\alpha}}.$$

So we can know that:

$$\mathbb{E}\left[\left(\frac{1}{\sqrt{n}}\sum_{t=1}^{T}\|\mathbf{v}_t\|^2\right)^{1-\alpha}\right] \leq 4C_3.$$

which indicate that

$$\mathbb{E}\left[\left(\frac{1}{T}\sum_{t=1}^{T}\|\mathbf{v}_t\|^2\right)^{1-\alpha}\right] \leq 4C_3\left(\frac{\sqrt{n}}{T}\right)^{1-\alpha}.$$

as well as

$$\mathbb{E}\left[\frac{1}{T}\sum_{t=1}^{T}\|\mathbf{v}_t\|\right] \leq (4C_3)^{\frac{1}{2(1-\alpha)}} \cdot \frac{n^{1/4}}{T^{1/2}}.$$

To finish the proof, we also have to show the following lemma.

**Lemma 8**

$$\mathbb{E}\left[\sum_{t=1}^{T}\|\nabla F(\mathbf{x}_t) - \mathbf{v}_t\|^2\right]$$

$$\leq \frac{\alpha\sqrt{n}}{1-\alpha}\left(\frac{14L^2}{1-2\alpha}\left(\frac{2-4\alpha}{(1-\alpha)L}\right)^{\frac{1-2\alpha}{1-\alpha}}\right)^{\frac{1-\alpha}{\alpha}} + \frac{n^{\frac{\alpha}{2}}L}{2}\mathbb{E}\left[\left(\sum_{t=1}^{T}\|\mathbf{v}_t\|^2\right)^{1-\alpha}\right].$$

**Proof 8** *According to the previous proof, we know that:*

$$\sum_{t=1}^{T}\mathbb{E}\left[\|\nabla F(\mathbf{x}_t) - \mathbf{v}_t\|^2\right]$$

$$\leq 2\beta\sum_{t=1}^{T}\mathbb{E}\left[\|\nabla F(\mathbf{x}_{t+1}) - \mathbf{z}_{t+1}\|^2\right] + 2L^2\sum_{t=1}^{T}\mathbb{E}\left[\frac{\eta_t^2}{\beta}\|\mathbf{v}_t\|^2\right] \leq 14nL^2\sum_{t=1}^{T}\mathbb{E}\left[\eta_t^2\|\mathbf{v}_t\|^2\right].$$

*Also, we can deduce that:*

$$14nL^2\sum_{t=1}^{T}\eta_t^2\|\mathbf{v}_t\|^2 = 14L^2n^\alpha\sum_{t=1}^{T}\frac{\|\mathbf{v}_t\|^2}{\left(\sum_{i=1}^{t}\|\mathbf{v}_i\|^2\right)^{2\alpha}}$$

$$\leq \frac{14L^2n^\alpha}{1-2\alpha}\left(\sum_{t=1}^{T}\|\mathbf{v}_t\|^2\right)^{1-2\alpha}$$

$$= \frac{14L^2n^\alpha}{1-2\alpha}\left(\frac{2-4\alpha}{(1-\alpha)n^{\frac{\alpha}{2}}L}\right)^{\frac{1-2\alpha}{(1-\alpha)}}\left(\frac{(1-\alpha)n^{\frac{\alpha}{2}}L}{2-4\alpha}\right)^{\frac{1-2\alpha}{1-\alpha}}\left(\sum_{t=1}^{T}\|\mathbf{v}_t\|^2\right)^{1-2\alpha}$$

$$\leq \frac{\alpha}{1-\alpha}\left(\frac{14L^2n^\alpha}{1-2\alpha}\left(\frac{2-4\alpha}{(1-\alpha)n^{\frac{\alpha}{2}}L}\right)^{\frac{1-2\alpha}{1-\alpha}}\right)^{\frac{1-\alpha}{\alpha}} + \frac{n^{\frac{\alpha}{2}}L}{2}\left(\sum_{t=1}^{T}\|\mathbf{v}_t\|^2\right)^{1-\alpha}$$

*where the last inequality employs Young's inequality, such that $xy \leq \frac{\alpha}{1-\alpha}x^{\frac{1-\alpha}{\alpha}} + \frac{1-2\alpha}{1-\alpha}y^{\frac{1-\alpha}{1-2\alpha}}$ for positive $x, y$.*

As a result, we can ensure that

$$
\begin{aligned}
\frac{1}{T} \sum_{t=1}^{T} \mathbb{E}\left[\|\nabla F(\mathbf{x}_t)\|\right] \leq & \frac{1}{T} \mathbb{E}\left[\sum_{t=1}^{T} \|\mathbf{v}_t\|\right] + \frac{1}{T}\left[\sum_{t=1}^{T} \|\nabla f(\mathbf{x}_t) - \mathbf{v}_t\|\right] \\
\leq & \frac{n^{1/4}}{T^{1/2}}\left(\left(\frac{14L^2}{1-2\alpha}\left(\frac{2-4\alpha}{(1-\alpha)L}\right)^{\frac{1-2\alpha}{1-\alpha}}\right)^{\frac{1-\alpha}{2\alpha}} + \sqrt{2C_3 L} + (4C_3)^{\frac{1}{2(1-\alpha)}}\right) \\
= & \mathcal{O}\left(\left(\Delta_F^{\frac{1}{2(1-\alpha)}} + L^{\frac{1}{2\alpha}}\right)\frac{n^{1/4}}{T^{1/2}}\right).
\end{aligned}
$$

# E  Proof of Theorem 5

The analysis is very similar to that of Theorem 4, and the difference only appears in Lemma 5. For this new method, we can also prove the same lemma:

**Lemma 9**

$$
2\beta \sum_{t=1}^{T} \mathbb{E}\left[\eta_t \|\nabla F(\mathbf{x}_{t+1}) - \mathbf{z}_{t+1}\|^2\right] \leq 12L^2 \sum_{t=1}^{T} \mathbb{E}\left[\frac{\eta_t^3}{\beta}\|\mathbf{v}_t\|^2\right].
$$

**Proof 9** *This time, we have $\mathbf{z}_{t+1} = \nabla f_{i_{t+1}}(\mathbf{x}_{t+1}) - \nabla f_{i_{t+1}}(\mathbf{x}_\tau) + \nabla F(\mathbf{x}_\tau)$. And we can know that:*

$$
\begin{aligned}
& \mathbb{E}_{i_{t+1}}\left[\eta_t \|\nabla F(\mathbf{x}_{t+1}) - \mathbf{z}_{t+1}\|^2\right] \\
= & \mathbb{E}_{i_{t+1}}\left[\eta_t \|\nabla F(\mathbf{x}_{t+1}) - \nabla f_{i_{t+1}}(\mathbf{x}_{t+1}) + \nabla f_{i_{t+1}}(\mathbf{x}_\tau) - \nabla F(\mathbf{x}_\tau)\|^2\right] \\
\leq & \mathbb{E}_{i_{t+1}}\left[\eta_t \|\nabla f_{i_{t+1}}(\mathbf{x}_{t+1}) - f_{i_{t+1}}(\mathbf{x}_\tau)\|^2\right] \\
\leq & \eta_t L^2 \|\mathbf{x}_{t+1} - \mathbf{x}_\tau\|^2 \\
\leq & \eta_t L^2 I \sum_{i=\tau}^{t} \|\mathbf{x}_{i+1} - \mathbf{x}_i\|^2 \\
\leq & \eta_t L^2 I \sum_{i=\tau}^{t} \eta_i^2 \|\mathbf{v}_i\|^2 \leq L^2 I \sum_{i=\tau}^{t} \eta_i^3 \|\mathbf{v}_i\|^2.
\end{aligned}
$$

*By summing up, we have*

$$
\begin{aligned}
& 2\beta \sum_{t=1}^{T} \mathbb{E}\left[\eta_t \|\nabla F(\mathbf{x}_{t+1}) - \mathbf{z}_{t+1}\|^2\right] \\
\leq & 2\beta \mathbb{E}\left[\sum_{t=1}^{T} L^2 I \sum_{i=\tau}^{t} \eta_i^3 \|\mathbf{v}_i\|^2\right] \leq 2\beta L^2 I^2 \mathbb{E}\left[\sum_{t=1}^{T} \eta_t^3 \|\mathbf{v}_t\|^2\right] \leq 2L^2 \mathbb{E}\left[\sum_{t=1}^{T} \frac{\eta_t^3}{\beta}\|\mathbf{v}_t\|^2\right].
\end{aligned}
$$

The other analysis is exactly the same as that of Theorem 4.

