# OpenReview forum: "Adaptive Variance Reduction for Stochastic Optimization under Weaker Assumptions"
_NeurIPS.cc/2024/Conference — NeurIPS 2024 poster_

### Official Review · Reviewer_4Ggd · 2024-07-10

**Soundness:** 4
**Presentation:** 4
**Contribution:** 4
**Rating:** 7
**Confidence:** 4

**Summary:**

This paper introduces novel adaptive variance reduction methods for stochastic optimization, building on the STORM technique. The proposed Ada-STORM method closes the $O(\log T)$ gap and achieves an optimal convergence rate of $O(T^{-1/3})$ for non-convex functions under assumptions weaker than previous approaches. Furthermore, the paper extends the method to stochastic compositional optimization, maintaining a similar $O(T^{-1/3})$ convergence rate. For non-convex finite-sum problems, the authors develop another innovative adaptive algorithm that attains the optimal $O(n^{1/4}T^{-1/2})$ rate. Numerical experiments effectively demonstrate the effectiveness of the proposed method.

**Strengths:**

1. The proposed Ada-STORM method overcomes major limitations in existing adaptive STORM methods. It does not require bounded gradient and function value assumptions, and achieves the optimal convergence rates without the additional $O(\log T)$ term. The problem investigated in the paper is challenging and the results are substantial.
2. The theoretical analysis is novel and easy to follow. Extending the method to stochastic compositional and finite-sum optimization problems illustrates its flexibility and potential impact across a wide range of optimization tasks.
3. The numerical experiments on various tasks (e.g., image classification, language modeling) validate the theoretical results and highlight the method's superior performance compared to other algorithms.

**Weaknesses:**

1. In Theorem 4, while the authors clearly state the convergence rate of the proposed method in terms of $T$, $\Delta_F$, and $L$, they only present the convergence of previous adaptive finite-sum methods in terms of $T$. A comparison of other important constants such as $\Delta_F$ and $L$ with existing methods could enhance the discussion.
2. The content in Lines 240 to 242, especially Algorithm 4, should be included in the main body of the paper rather than being delayed to the Appendix.
3. Some typos in the paper:
- Line 145: $(t>2)$ should be $(t \geq 2)$.
- Line 516, the first equality in Proof 7: $L^2$ should be $L$.

**Questions:**

1. Regarding Line 401, could the authors explain why the first inequality holds?
2. In Theorem 4, is the dependency on $\Delta_F$ and $L$ better or worse than that of previous methods?

**Limitations:**

This paper does not present any negative societal impacts.

---

> ### Author Rebuttal · Authors · 2024-08-06
>
> Thank you very much for your constructive comments!
>
> ---
>
> **Q1:** In Theorem 4, while the authors clearly state the convergence rate of the proposed method in terms of $T$, $\Delta_F$, and $L$, they only present the convergence of previous adaptive finite-sum methods in terms of $T$. A comparison of other important constants such as $\Delta_F$ and $L$ with existing methods would enhance the discussion.
>
> **A1:** According to Theorem 1 of the original paper [Kavis et al., 2022], the convergence rate of previous adaptive finite-sum methods is $\mathcal{O}\left(n^{1 / 4} T^{-1/2} \left(L^2 + \Delta_F \right) \cdot \log \left(1+n T L\right)\right)$. In contrast, our convergence rate obtained in Theorem 4 is $\mathcal{O}\left(n^{1 / 4} T^{-1/2} \left(L^{\frac{1}{2\alpha}} + \Delta_F^{\frac{1}{2(1-\alpha)}} \right) \right)$, where $\alpha$ can be any constant within $(0,1/3)$. For example, if we set $\alpha = 1/4$, the rate reduces to $\mathcal{O}\left(n^{1 / 4} T^{-1/2} \left(L^2 + \Delta_F^{{2}/{3}} \right) \right)$, which is better than the previous convergence rate.
>
> ---
>
> **Q2:** The content in Lines 240 to 242, especially Algorithm 4, should be included in the main body of the paper rather than being delayed to the Appendix.
>
> **A2:** Thank you for your suggestion! We will move Algorithm 4 to the main body of the paper and add more discussion about this algorithm.
>
> ---
>
> **Q3:** There are some typos in the paper.
>
> **A3:** Thank you for catching the typos. We will correct them in the revised version.
>
> ---
>
> **Q4:** Regarding Line 401, could the authors explain why the first inequality holds?
>
> **A4:** To address your question, we provide a more detailed analysis here:
>
>   \begin{align}
>         & \mathbb{E}\_{\xi_{t+1}} \left[ ||(1-\beta)(\mathbf{v}_t - \nabla f(\mathbf{x}_t)) + \left(\nabla f(\mathbf{x}_t)-\nabla f(\mathbf{x}\_{t+1}) + \nabla f(\mathbf{x}\_{t+1};\xi\_{t+1}) - \nabla f(\mathbf{x}\_{t};\xi\_{t+1}) \right)   + \beta\left(\nabla f(\mathbf{x}\_{t};\xi\_{t+1}) - \nabla f(\mathbf{x}\_{t})  \right) ||^2 \right]\\\\
>          \leq & \mathbb{E}\_{\xi\_{t+1}}\left[(1-\beta)^2||\mathbf{v}\_{t} - \nabla f(\mathbf{x}\_{t})||^2\right]  + \mathbb{E}\_{\xi\_{t+1}}\left[|| \left(\nabla f(\mathbf{x}\_{t})-\nabla f(\mathbf{x}\_{t+1}) + \nabla f(\mathbf{x}\_{t+1};\xi\_{t+1}) - \nabla f(\mathbf{x}\_{t};\xi\_{t+1}) \right)   + \beta\left(\nabla f(\mathbf{x}\_{t};\xi\_{t+1}) - \nabla f(\mathbf{x}\_{t})  \right) ||^2 \right] \\\\
>         \leq & \mathbb{E}\_{\xi\_{t+1}}\left[(1-\beta)^2||\mathbf{v}\_{t} - \nabla f(\mathbf{x}\_{t})||^2\right] + \mathbb{E}\_{\xi\_{t+1}}\left[2|| \nabla f(\mathbf{x}\_{t})-\nabla f(\mathbf{x}\_{t+1}) + \nabla f(\mathbf{x}\_{t+1};\xi\_{t+1}) - \nabla f(\mathbf{x}\_{t};\xi\_{t+1}) ||^2  + 2|| \beta\left(\nabla f(\mathbf{x}\_{t};\xi\_{t+1}) - \nabla f(\mathbf{x}\_{t})  \right)||^2\right] \\\\
>         \leq & (1-\beta)^2 \mathbb{E}\_{\xi\_{t+1}}\left[||\mathbf{v}\_{t} - \nabla f(\mathbf{x}\_{t})||^2\right] +2 \mathbb{E}\_{\xi\_{t+1}}\left[||\nabla f(\mathbf{x}\_{t+1};\xi\_{t+1}) - \nabla f(\mathbf{x}\_{t};\xi\_{t+1}) ||^2\right] + 2\beta^2 \mathbb{E}\_{\xi\_{t+1}}\left[||\nabla f(\mathbf{x}\_{t};\xi\_{t+1}) - \nabla f(\mathbf{x}\_{t})||^2\right]
>     \end{align}
> where the first inequality is due to the fact that
> \begin{align}
>      \mathbb{E}\_{\xi\_{t+1}}\left[ \nabla f(\mathbf{x}\_{t})-\nabla f(\mathbf{x}\_{t+1}) + \nabla f(\mathbf{x}\_{t+1};\xi\_{t+1}) - \nabla f(\mathbf{x}\_{t};\xi\_{t+1})    + \beta\left(\nabla f(\mathbf{x}\_{t};\xi\_{t+1}) - \nabla f(\mathbf{x}\_{t})  \right)  \right] = 0,
> \end{align}
> and the last inequality is due to:
> \begin{align}
> & \mathbb{E}\_{\xi\_{t+1}}\left[|| \nabla f(\mathbf{x}\_{t})-\nabla f(\mathbf{x}\_{t+1}) + \nabla f(\mathbf{x}\_{t+1};\xi\_{t+1}) - \nabla f(\mathbf{x}\_{t};\xi\_{t+1})  ||^2\right] \\\\
> \leq & \mathbb{E}\_{\xi\_{t+1}}\left[|| \nabla f(\mathbf{x}\_{t})-\nabla f(\mathbf{x}\_{t+1}) ||^2\right] + \mathbb{E}\_{\xi\_{t+1}}\left[||\nabla f(\mathbf{x}\_{t+1};\xi\_{t+1}) - \nabla f(\mathbf{x}\_{t};\xi\_{t+1}) ||^2\right]  +2 \mathbb{E}\_{\xi\_{t+1}} \left\langle \nabla f(\mathbf{x}\_{t})-\nabla f(\mathbf{x}\_{t+1}), \nabla f(\mathbf{x}\_{t+1};\xi\_{t+1}) - \nabla f(\mathbf{x}\_{t};\xi\_{t+1})  \right\rangle\\\\
>  \leq & \mathbb{E}\_{\xi\_{t+1}}\left[|| \nabla f(\mathbf{x}\_{t+1};\xi\_{t+1}) - \nabla f(\mathbf{x}\_{t};\xi\_{t+1})  ||^2\right] - \mathbb{E}\_{\xi\_{t+1}}\left[|| \nabla f(\mathbf{x}\_{t})-\nabla f(\mathbf{x}\_{t+1})  ||^2\right]\\\\
> \leq & \mathbb{E}\_{\xi\_{t+1}}\left[||\nabla f(\mathbf{x}\_{t+1};\xi\_{t+1}) - \nabla f(\mathbf{x}\_{t};\xi\_{t+1})||^2\right]
> \end{align}
>
> ---
>
> **Q5:** In Theorem 4, is the dependency on $\Delta_F$ and $L$ better or worse than that of previous methods?
>
> **A5:** As discussed in A1, our dependency on $\Delta_F$ and $L$ is better than previous methods.
>
> ---
>
> **Reference:**
>
> Kavis et al. Adaptive stochastic variance reduction for non-convex finite-sum minimization. NeurIPS, 2022.

---

> > ### Comment · Reviewer_4Ggd · 2024-08-09
> >
> > Thank the authors for the rely. It addresses all my questions and concerns. I would keep my score.

---

### Official Review · Reviewer_2239 · 2024-07-10

**Soundness:** 3
**Presentation:** 3
**Contribution:** 4
**Rating:** 7
**Confidence:** 2

**Summary:**

This paper proposes a novel adaptive STORM method that achieves an optimal convergence rate of $O(T^{-3})$ for nonconvex stochastic optimization, which requires weaker assumptions and attains the optimal convergence rate without the additional $O(\log T)$ term.

**Strengths:**

For stochastic non-convex optimization, this paper achieves the optimal convergence rate under more relaxed assumptions, which does not require the bounded function values or the bounded gradients and does not include the additional $O(\log T)$ term in the convergence rate. The proposed technique has also been extended to stochastic compositional optimization with the same optimal rate. For non-convex finite-sum optimization, this papaer further improve the adaptive algorithm to attain an optimal convergence rate, which outperforms the previous result by $O(\log(nT))$ factor.

**Weaknesses:**

Although this paper is theoretically sound in general, there are still some questions need to be discussed:
1.	The experimental part is relatively limited. The proposed algorithm is parameter free, so it is best to provide some experimental results to demonstrate whether these compared algorithms are sensitive to parameter changes.
2.	In theory, the output {x_ \tau} of the proposed algorithms is randomly selected from {1,..,T}. Is the output in the experiment randomly selected?

**Questions:**

1.	The experimental part is relatively limited. The proposed algorithm is parameter free, so it is best to provide some experimental results to demonstrate whether these compared algorithms are sensitive to parameter changes.
2.	In theory, the output {x_ \tau} of the proposed algorithms is randomly selected from {1,..,T}. Is the output in the experiment randomly selected?

**Limitations:**

The authors discussed the limitations and there is no negative societal impact about this work.

---

> ### Author Rebuttal · Authors · 2024-08-06
>
> Thank you very much for your constructive comments and suggestions.
>
> ---
>
> **Q1:** The experimental part is relatively limited. The proposed algorithm is parameter-free, so it is best to provide some experimental results to demonstrate whether these compared algorithms are sensitive to parameter changes.
>
>
> **A1:** Following your suggestion, we have included additional experiments to assess whether the original STORM method is sensitive to changes in hyper-parameters. Specifically, we tested the initial learning rate from the set {$0.01, 0.1,1,10,50$}. The results are reported in the **Global Response (Figure 1)**, and we observe that the STORM method is sensitive to changes in hyper-parameters.
>
> ---
>
> **Q2:** In theory, the output {$x_{\tau}$} of the proposed algorithms is randomly selected from {$1,..,T$}. Is the output in the experiment randomly selected?
>
> **A2:** In the experiment, we simply use the output of the last iteration, i.e., $\mathbf{x}_{\tau} = \mathbf{x}_T$. It is common practice to select the output randomly from {$1,\cdots, T$} in theoretical analysis but use the output of the last iteration in practical experiments [Johnson et al., 2013, Cutkosky et al., 2019]. This is also the case for the original STORM paper. Specifically, in Line 14 of the STORM algorithm [Cutkosky et al., 2019], they state: "Choose $\hat{\mathbf{x}}$ uniformly at random from {$\mathbf{x}_1, \cdots , \mathbf{x}_T$} . (In practice, set $\hat{\mathbf{x}} = \mathbf{x}_T $)."
>
> ---
>
> **References:**
>
> Johnson et al. Accelerating stochastic gradient descent using predictive variance reduction. NeurIPS, 2013
>
> Cutkosky et al. Momentum-based variance reduction in non-convex SGD. NeurIPS, 2019.

---

> > ### Comment · Reviewer_2239 · 2024-08-10
> >
> > Thanks for the response and I have no further questions.

---

### Official Review · Reviewer_qB9i · 2024-07-10

**Soundness:** 3
**Presentation:** 3
**Contribution:** 3
**Rating:** 7
**Confidence:** 3

**Summary:**

This paper studies non-convex stochastic optimization under the assumption of mean-square smoothness. It introduces, Ada-STORM, a variant of the STORM algorithm, which achieves the optimal rate $O(T^{-1/3})$. Unlike vanilla STORM, Ada-STORM eliminates the $O(\log T)$ factor and does not require the Lipschitz assumption. The algorithm's success hinges on a carefully designed adaptive learning rate schedule. Instead of the standard AdaGrad learning rate $\eta_t \approx (\sum_{i=1}^t \\|g_i\\|^2)^{-1/3}$ used in STORM, Ada-STORM employs $\eta_t = \min\\{T^{-1/3}, T^{-(1-\alpha)/3}(\sum_{i=1}^t \\|v_i\\|^2)^{-\alpha}\\}$ where $\alpha < 1/3$ and $v_t$ is the STORM update.

**Strengths:**

The proposed algorithm, Ada-STORM, achieves the optimal rate $O(T^{-1/3})$ without the log factor. Unlike STORM+ which also eliminates the log factor, this algorithm does not assume $f$ to be Lipschitz and bounded above. This improvement represents a significant contribution. Moreover, the required modification is straightforward: it involves merely adjusting the learning rate scheduler $\eta_t$ and the momentum factor $\beta_t$. I particularly enjoy the constant momentum constant $\beta_t\equiv T^{-2/3}$ rather than time-varying and dependent on $\eta_t$.

In addition, the proposed learning rate scheduler is of interest in its own right. The paper extends the application of this learning rate technique to related problems, achieving optimal rates in composite optimization and finite-sum optimization (ERM). Finally, empirical experiments also validate Ada-STORM's performance, which matches or outperforms existing STORM variants and other state-of-the-art optimizers.

**Weaknesses:**

One major concern pertains to the technical correctness of the proof of Theorem 1, in particular from line 440-444. I don't immediately see how $\mathbb{E}[\sum_{t=1}^{s-1} \\|v_t\\|^2] \le C_0T^{1/3}$ implies $\mathbb{E}[\frac{1}{T}\sum_{t=1}^{s-1}\\|v_t\\|] \le \sqrt{C_0}T^{-1/3}$, and similarly how the equation in line 441 implies line 442. My concern arise from the fact that $s$ is a random variable dependent on the entire history $\mathcal{H}_T$. As a result, applying Jensen's inequality like $\mathbb{E}[\frac{1}{T}\sum\_{t=1}^s\\|v_t\\|] \le \sqrt{\mathbb{E}[\frac{1}{T}\sum\_{t=1}^s\\|v_t\\|^2]}$ seems to be inappropriate when $s$ is a random variable dependent $v_t$'s. It's likely that I miss something, and I encourage the authors to elaborate this part of the proof to make it clear.

Another concern is about the doubling trick. In each stage $k$, the guarantee is
$$\frac{1}{2^{k-1}} \sum_{t=2^{k-1}}^{2^k} \mathbb{E}\\|\nabla f(x_t)\\| \lesssim \frac{\Delta_f^{3/4}+O(1)}{(2^{k-1})^{1/3}}.$$
Since the algorithm is not resetting $x_t=x_0$ in the new stage, the parameter $\Delta_f$ is not constant. In particular, in stage $k$, $\Delta_f = \mathbb{E}[f(x_{2^{k-1}}) - \inf f(x)]$. Consequently, the doubling trick implicitly assumes $f$ has bounded function value in order to bound $\Delta_f$ in all stages.

If the authors can address my major concerns, I am inclined to revise my score upwards, given the otherwise strong results of the paper.

**Questions:**

- Is there a specific reason why the authors prefer $\alpha$ as large as possible in the trade-off of $\Delta^{1/2(1-\alpha)} + \sigma^{1/(1-\alpha)} + L^{1/2\alpha}$. Is it due to the assumption (or some empirical observation) that the smoothness constant $L$ is usually the dominating parameter?

- In the experiments, did the authors apply any learning rate scheduler (e.g., linear decay or cosine decay) on top of the adaptive learning rate? Although somewhat tangential, it would be interesting to compare the new adaptive scheduler with other popular schedulers.

---

> ### Author Rebuttal · Authors · 2024-08-06
>
> Thanks for your constructive comments, and we will revise our paper accordingly! We have addressed the major concerns about the proof as outlined below. We sincerely hope that the reviewer can examine them and reevaluate our results.
>
> ---
>
> **Q1:** One major concern pertains to the technical correctness of the proof of Theorem 1, in particular from line 440-444. I don't immediately see how $\mathbb{E} [\sum_{t=1}^{s-1} ||v_t||^2] \leq C_0 T^{1/3}$ implies $\mathbb{E}[\frac{1}{T}\sum_{t=1}^{s-1}||v_t||] \leq \sqrt{C_0}T^{-1/3}$, and similarly how the equation in line 441 implies line 442. My concern arise from the fact that $s$ is a random variable dependent on the entire history $\mathcal{H}\_T$. As a result, applying Jensen's inequality like $\mathbb{E}[\frac{1}{T}\sum_{t=1}^s||v_t||] \le \sqrt{\mathbb{E}[\frac{1}{T}\sum_{t=1}^s||v_t||^2]}$ seems inappropriate when $s$ is a random variable dependent on $v_t$'s.
>
> **A1:** Sorry for omitting the details. We provide a more detailed analysis as follows, which will be added to our revised paper. First, we use the inequality $\mathbb{E} [X] \leq \sqrt{\mathbb{E}[X^2]}$, which is due to Jensen's inequality. By setting $X = \frac{1}{T} \sum_{t=1}^s||v_t||$, we obtain $\mathbb{E} \left[ \frac{1}{T}\sum_{t=1}^s||v_t|| \right] \leq \sqrt{ \mathbb{E}\left[\left(\frac{1}{T}\sum_{t=1}^s||v_t||\right)^2\right]}$. We think the above inequalities hold regardless of the fact that $s$ is a random variable dependent on $v_t$'s. Furthermore, even without using Jensen's inequality, we can still prove the same result as follows:
> \begin{align}
>     0 \leq& \mathbb{E} \left[ \left( \frac{1}{T}\sum_{t=1}^s||v_t||  -  \mathbb{E}\left[\frac{1}{T}\sum_{t=1}^s||v_t||\right] \right)^2\right]\\\\
>     =& \mathbb{E} \left[  \left(\frac{1}{T}\sum_{t=1}^s||v_t||\right)^2 \right]  +\left( \mathbb{E}\left[\frac{1}{T}\sum_{t=1}^s||v_t||\right] \right)^2 - 2\mathbb{E}\left[\left\langle \frac{1}{T}\sum_{t=1}^s||v_t||, \mathbb{E}\left[\frac{1}{T}\sum_{t=1}^s||v_t||\right]
>  \right\rangle\right] \\\\
>  =& \mathbb{E} \left[  \left(\frac{1}{T}\sum_{t=1}^s||v_t||\right)^2 \right]  -\left( \mathbb{E}\left[\frac{1}{T}\sum_{t=1}^s||v_t||\right] \right)^2,
> \end{align}
> which indicates that $\mathbb{E}\left[\frac{1}{T}\sum_{t=1}^s||v_t||\right]  \leq \sqrt{\mathbb{E} \left[  \left(\frac{1}{T}\sum_{t=1}^s||v_t||\right)^2 \right]} $. We can complete the proof by noting that:
> \begin{align*}
>     \mathbb{E}\left[\frac{1}{T}\sum_{t=1}^s||v_t||\right]  &\leq \sqrt{\mathbb{E} \left[  \left(\frac{1}{T}\sum_{t=1}^s||v_t||\right)^2 \right]} = \sqrt{\mathbb{E} \left[ \frac{1}{T^2} \left(\sum_{t=1}^s||v_t||\right)^2 \right]} \\\\
>     &\leq \sqrt{\mathbb{E} \left[ \frac{s}{T^2} \sum_{t=1}^s||v_t||^2 \right]} \leq \sqrt{\mathbb{E} \left[ \frac{1}{T} \sum_{t=1}^s||v_t||^2 \right]} \\\\
>     &\leq  \sqrt{ \frac{1}{T} C_0 T^{1/3}} = \sqrt{C_0 }T^{-1/3}.
> \end{align*}
>
> ---
>
> **Q2:** Another concern is about the doubling trick. In each stage $k$, the guarantee is $\frac{1}{2^{k-1}} \sum_{t=2^{k-1}}^{2^k} \mathbb{E}||\nabla f(x_t)|| \lesssim \frac{\Delta_f^{3/4}+O(1)}{(2^{k-1})^{1/3}}$. Since the algorithm is not resetting $x_t=x_0$ in the new stage, the parameter $\Delta_f$ is not constant.
>
> **A2:** We apologize for the confusion. In the analysis, we actually reset $x_t=x_0$ at the beginning of each new stage, and we will clearly clarify it in the revised paper. This reset ensures that the initial gap can always be bounded.  Since the iteration number doubles with each new stage, we can guarantee that the last complete stage has at least $T/4$ iterations. According to the analysis of Theorem 1, running Algorithm 1 for $T/4$ iterations leads to the following guarantee:
> \begin{align}
>     \mathbb{E}\left[ ||\nabla f(\textbf{x}_{\tau})||^2 \right] \leq \mathcal{O}\left(\frac{\Delta_f^{\frac{1}{2(1-\alpha)}}+\sigma^{\frac{1}{1-\alpha}} + L^{\frac{1}{2\alpha}}}{{(T/4)}^{1/3}}\right)=\mathcal{O}\left(\frac{\Delta_f^{\frac{1}{2(1-\alpha)}}+\sigma^{\frac{1}{1-\alpha}} + L^{\frac{1}{2\alpha}}}{{T}^{1/3}}\right).
> \end{align}
>
> ---
>
> **Q3:** Is there a specific reason why the authors prefer $\alpha$ as large as possible in the trade-off of $\Delta^{1/2(1-\alpha)} + \sigma^{1/(1-\alpha)} + L^{1/2\alpha}$. Is it due to the assumption (or some empirical observation) that the smoothness constant is usually the dominating parameter?
>
> **A3:** No, there is no specific reason to prefer larger $\alpha$ in the trade-off $\Delta^{1/2(1-\alpha)} + \sigma^{1/(1-\alpha)} + L^{1/2\alpha}$. Whether the smoothness constant $L$ or the parameters $\Delta$ and $\sigma$ dominate usually depends on the specific problem. To avoid misunderstandings, We will change our expression from “larger $\alpha$ leads to better dependence on parameter $L$” to “larger $\alpha$ leads to better dependence on the parameter $L$ and worse reliance on parameters $\Delta$ and $\sigma$”.
>
> ---
>
> **Q4:** In the experiments, did the authors apply any learning rate scheduler (e.g., linear decay or cosine decay) on top of the adaptive learning rate? Although somewhat tangential, it would be interesting to compare the new adaptive scheduler with other popular schedulers.
>
> **A4:** We did not apply learning rate scheduler on top of the adaptive learning rate in the experiments. Based on your suggestion, we conducted additional experiments comparing the STORM method with linear decay and cosine decay schedulers. The results are shown in **Global Response (Figure 2)**, which indicate that the performances of STORM with linear decay and cosine decay are still worse than our proposed method in terms of testing accuracy.

---

> > ### Comment · Reviewer_qB9i · 2024-08-08
> >
> > Thank the authors for the response.
> >
> > Regarding the technical details in Theorem 1, sorry for not making my point clear initially. I wasn't questioning the validity of Jensen's inequality. Rather, I was unclear about how to apply Jensen's inequality properly in this context. I appreciate that the authors have now clarified this.
> >
> > Regarding the doubling trick, the authors resolved my misunderstanding. Please clarify this in later drafts.
> >
> > Regarding the additional experiments, I thank the authors for the extra efforts. Although tested on the rather simple dataset CIFAR-10, the additional results that the new adaptive rate still outperforms Storm with other learning rate schedulers (e.g. linear decay, cosine decay) strongly supports the validity of Ada-Storm. If time is allowed, I'd suggest the authors to repeat the experiments on larger tasks such as LLMs in the future version, and it will be an encouraging result if Ada-Storm also outperforms other optimizers on more complicated tasks.
> >
> > Overall, the authors have addressed my major concerns. Given its concrete theoretical results and encouraging empirical results, I revised my score accordingly.

---

> > > ### Author Response · Authors · 2024-08-09
> > >
> > > Thank you very much for your kind reply! We will improve our paper according to your constructive reviews.
> > >
> > > Best regards,
> > >
> > > Authors

---

### Official Review · Reviewer_G4tN · 2024-07-15

**Soundness:** 3
**Presentation:** 2
**Contribution:** 2
**Rating:** 6
**Confidence:** 3

**Summary:**

This paper studies adaptive variants of STORM, a variance reduction technique proposed by Cutkosky and Orabona (2019), for nonconvex stochastic minimization problems. Through introducing a novel adaptive parameter and step size tuning method, the authors aim to remove the bounded gradients and bounded function values assumptions in existing literatures while achieving the optimal convergence rate without an additional penalty of $O(\log T)$ term due to adaptiveness. Based on the same techniques, they further propose Compositional STORM for compositional functions and STORM for finite-sum problems. They support their theoretical claims with numerical experiments on image classification and small-scale language modelling tasks.

**Strengths:**

1. Their proposed method can achieve optimal convergence rate of $O(T^{-1/3})$ under more relaxed assumptions, namely by removing the need for bounded function values and bounded gradients. Their method does not incur a $\log(T)$ term penalty like in other adaptive methods (Liu et al., 2022).
2. They show that their learning rate design and analysis technique can be extended to stochastic compositional problems under further assumptions, namely Lipschitz continuity.
3. Although rather unsurprising, they show that when the finite-sum structure is present, their method can be adapted to and showed improved convergence rates of a log factor over existing methods.

**Weaknesses:**

1. My understanding of the original non-adaptive STORM proposed by Cutkosky and Orabona (2019) does not have a log factor in their convergence rate when noise is present. Specifically their convergence rates in expectation is $O(\log T / T^{1/2} + \sigma^{1/3} / T^{1/3})$, thus the $\log T$ term is dominated and should not appear in your big-O convergence rates in many places of the paper (e.g., Line 28, and Table 1). Can you please explain and check your comparisons?

2. I am not sure about the significance of removing the bounded gradients and bounded function values assumption in practice. Cutkosky and Orabona (2019) proposed an adaptive $G_t$ approach to remove the bounded gradients assumption in their original paper, while bounded function values seems to be a very mild assumption under smoothness conditions when the final bounds all depend on the initial function value gap. Thus, I do not agree with the authors claiming them to be "strong assumptions". It would be useful if the authors can provide examples to justify why these assumptions can be problematic in practice.

3. The authors should be clearer that their extension to stochastic compositional optimization is not exactly under weaker assumptions. They require standard Lipschitz continuity assumptions which have been standard in the literature, which in essence is same as bounded gradients.

Overall I think this paper has potential but may benefit from addressing the above points.

**Questions:**

See weaknesses.

**Limitations:**

Yes.

---

> ### Author Rebuttal · Authors · 2024-08-06
>
> Thank you very much for the constructive review!
>
> ---
>
> **Q1:** My understanding of the original non-adaptive STORM proposed by Cutkosky and Orabona (2019) does not have a log factor in their convergence rate when noise is present. Specifically their convergence rates in expectation is $O(\log T / T^{1/2} + \sigma^{1/3} / T^{1/3})$, thus the $\log T$ term should not appear in your big-O convergence rates.
>
> **A1:** Although the convergence rate presented in the paper [Cutkosky and Orabona, 2019] is indeed $O(\log T / T^{1/2} + \sigma^{1/3} / T^{1/3})$, it is not accurate after checking the proofs in detail. Specifically, at the end of Section 5 of their paper, they claim that
> \begin{align}
> \mathbb{E}\left[\sum\_{t=1}^{T} \frac{||\nabla F\left(\mathbf{x}\_{t}\right)||}{T}\right] \leq \frac{\sqrt{2 M}\left(w+2 T \sigma^{2}\right)^{1 / 6}+2 M^{3 / 4}}{\sqrt{T}} \leq \frac{w^{1 / 6} \sqrt{2 M}+2 M^{3 / 4}}{\sqrt{T}}+\frac{2 \sigma^{1 / 3}}{T^{1 / 3}} ,
> \end{align}
> where they utilize $(a+b)^{1 / 3} \leq a^{1 / 3}+b^{1 / 3}$ in the last inequality and $M$ is a complex term containing the $\log T$ term. However, they neglect the $\sqrt{M}$ factor in the last term, which should be:
> \begin{align*}
> \mathbb{E}\left[\sum_{t=1}^T \frac{||\nabla F\left(\boldsymbol{x}_{t}\right)||}{T}\right] \leq \frac{\sqrt{2 M}\left(w+2 T \sigma^{2}\right)^{1 / 6}+2 M^{3 / 4}}{\sqrt{T}}\leq \frac{w^{1 / 6} \sqrt{2 M}+2 M^{3/4}}{\sqrt{T}}+\frac{2 \sigma^{1 / 3} \color{red}{\sqrt{M}}}{T^{1 / 3}} .
> \end{align*}
> As a result, the $\log T$ term should appear in the big-O convergence rates.
>
> ---
>
> **Q2:** I am not sure about the significance of removing the bounded gradients and bounded function values assumption in practice. Cutkosky and Orabona (2019) proposed an approach to remove the bounded gradients assumption in their original paper, while bounded function values seems to be a very mild assumption under smoothness conditions when the final bounds all depend on the initial function value gap. Thus, I do not agree with the authors claiming them to be "strong assumptions". It would be useful if the authors can provide examples to justify why these assumptions can be problematic in practice.
>
> **A2:** Thank you for your valuable comment! First, although Cutkosky and Orabona (2019) remove the bounded gradient assumption, their newly proposed algorithm introduces two new drawbacks:
> (1) As stated in their paper, this method requires knowledge of the true value of gradient variance $\sigma$, which is impractical; (2) The learning rate of this new algorithm is deterministic, i.e., $\eta_t = \frac{k}{(w+\sigma^2 t)^{1/3}}$, which cannot adjust according to the stochastic gradient anymore. Second, the bounded gradient assumption is still required in the STORM+ method [Levy et al., 2021], and it cannot be removed using the same technique as Cutkosky and Orabona (2019) due to the more complex analysis involved in STORM+.
>
> Regarding the bounded function value assumption, we believe it is indeed a strong assumption. The most commonly used linear and quadratic functions are both smooth but have unbounded function values. Furthermore, this assumption introduces an extra term related to the function value upper bound in the convergence rate. Specifically, STORM+ [Levy et al., 2021] achieves a convergence rate of $\mathcal{O}\left( \left(B^{3/4} + L^{3/2} \right) \sigma^{1/3} T^{-1/3}\right)$, where $B$ is the upper bound of the function value. The overall convergence rate can be significantly impacted when $B$ is large.
>
> ---
>
> **Q3:** The authors should be clearer that their extension to stochastic compositional optimization is not exactly under weaker assumptions. They require standard Lipschitz continuity assumptions which have been standard in the literature, which in essence is same as bounded gradients.
>
>
> **A3:** We agree that in stochastic compositional optimization, the required Lipschitz continuity is equivalent to the bounded gradient assumption. However, this assumption is inherently introduced by the compositional optimization itself rather than by our adaptive techniques. Note that the bounded gradient assumption is essential and widely required in the literature for stochastic compositional optimization [Wang et al., 2017, Yuan et al., 2019, Zhang et al., 2019]. Additionally, we would like to emphasize that our method does not require the bounded function value assumption, which would be needed if we attempt to apply the same techniques from STORM+ [Levy et al., 2021] to stochastic compositional optimization. We will incorporate these clarifications into our revised paper to make it more clear.
>
> ---
>
> **References:**
>
> A. Cutkosky and F. Orabona. Momentum-based variance reduction in non-convex SGD. NeurIPS, 2019.
>
> Levy et al. STORM+: Fully adaptive SGD with recursive momentum for nonconvex optimization. NeurIPS, 2021.
>
> Wang et al. Accelerating stochastic composition optimization. JMLR, 2017.
>
> Yuan et al. Efficient smooth non-convex stochastic compositional optimization via stochastic recursive gradient descent. NeurIPS, 2019.
>
> Zhang et al. A stochastic composite gradient method with incremental variance reduction. NeurIPS, 2019.

---

> ### Comment · Reviewer_G4tN · 2024-08-11
>
> Thank you for detailed explanation, and upon further checking, I agree with the authors' claims regarding my W1 and W2. As such, I am increasing my score from 5 to 6.

---

> > ### Author Response · Authors · 2024-08-11
> >
> > Many thanks for your kind response and supports! We will revise our paper according to your constructive suggestions.

---

### Author Rebuttal · Authors · 2024-08-06

## **Global Response** ##

---

In response to the request of reviewers, we provide additional experimental results in this part.

**Figure 1:** According to the suggestion of Reviewer 2239, we provide the performance of the STORM method with different initial learning rates. Specifically, we tested learning rates from the set {$0.01, 0.1, 1, 10, 50$}. The results demonstrate that the STORM method is sensitive to the choice of the hyper-parameters.

**Figure 2:** As requested by Reviewer qB9i, we include experiments on the STORM method with linear decay and cosine decay schedulers for comparison. The results indicate that the performances of STORM with linear decay and cosine decay are still worse than our Ada-STORM method in terms of testing accuracy.

---

### Decision · Program_Chairs · 2024-09-25

**Decision:**

Accept (poster)

**Comment:**

This is a paper about stochastic optimization of nonconvex functions. The main results show that the STORM technique, which achieves variance reduction, can be shown to achieve optimal rates without overly restrictive assumptions (such as bounded gradients or bounded function values). In particular, log terms from previous work are avoided. The authors adapt the same techniques to compositional and finite-sum settings, also achieving state-of-the-art results (albeit under stronger assumptions).

The reviewers' scores (6,7,7,7) strongly suggest acceptance, and I concur with their assessment. Although perhaps technical, it seems that the paper presents a cogent message and interesting new ideas. The discussion period was crucial in clarifying some misunderstandings and difficulties with the proofs, but the authors seem clear on which points of their manuscript need revision. Overall, I am happy to suggest that this paper be accepted to NeurIPS.